# SILENCER: PRUNING-AWARE BACKDOOR DEFENSE FOR DECENTRALIZED FEDERATED LEARNING

## ABSTRACT

Decentralized Federated Learning (DFL) with gossip protocol has to cope with a much larger attack surface under backdoor attacks, because by adopting aggregation without central coordination, a small percentage of adversaries in DFL may directly gossip the poisoned model updates to their neighbors, and subsequently broadcast the poisoning effect to the entire peer-to-peer (P2P) network. By examining backdoor attacks in DFL, we discover an exciting phenomenon that the poisoned parameters on adversaries have distinct patterns on their diagonal of empirical Fisher information (FI). Next, we show that such invariant FI patterns can be utilized to cure the poisoned models through effective model pruning. Unfortunately, we also observe an unignorable downgrade of benign accuracy of models when applying the naive FI-based pruning. To attenuate the negative impact of FI-based pruning, we present SILENCER, a *dynamic two-stage model pruning scheme* with robustness and accuracy as dual goals. At the first stage, SILENCER employs a FI-based parameter pruning/reclamation process during per-client local training. Each client utilizes a sparse surrogate model for local training, in order to be aware and reduce the negative impact of the second stage. At the second stage, SILENCER performs consensus filtering to remove dummy/poisoned parameters from the global model, and recover a benign sparse core model for deployment. Extensive experiments, conducted with three representative DFL settings, demonstrate that SILENCER *consistently* outperforms existing defenses by a large margin. Our code is available at `https://anonymous.4open.science/r/Silencer-8F08/`.

## 1 INTRODUCTION

The success of machine learning cannot be achieved without accessing a massive amount of training data. Although most of the training data to date are publicly available, a growing number of downstream learning tasks will use proprietary or sensitive data. As a surrogate solution to large-scale centralized training, Decentralized Federated Learning (DFL) is gaining increased attention in recent years (Roy et al., 2019). Similar to conventional federated learning (FL), clients in DFL keep their proprietary data locally and only share their model updates with the server(s). However, unlike conventional FL, aggregation in DFL is performed through fully decentralized coordination (Stoica et al., 2003; Maymounkov & Mazieres, 2002) over a peer to peer (P2P) network. By employing DFL with randomized gossip protocol (Yang et al., 2022), no centralized entity will have full access to the local training models of all participating clients, eliminating some known threats in conventional FL (See Appendix A.1).

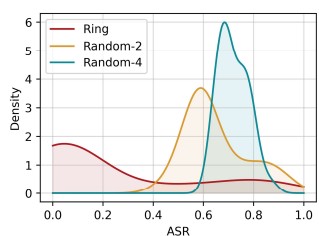

Figure 1: Kernel density of client models under three topology.

However, DFL also opens the door to new attack surfaces, making it more vulnerable to backdoor attack (Bagdasaryan et al., 2020). Even using a small percentage of attacker, the attackers can poison the model of potentially every peer in the system. Figure 1 reports a case study by poisoning the DFL network with 10% of attackers. We launch the attack under three different types of decentralized topology: Ring, random with 2 neighbors, and random with 4 neighbors. We measure the kernel density in terms of the Attack Success ratio (ASR) of each client's model. Our results show that DFL systems in all three settings suffer varying extents of poisoning effects. The random topology with a larger number of neighbors tends to be racked with a more severe poisoning effect with higher ASR.

Consider random-4, most of the clients' models experience ASR of over $70\%$ compared to the ASR of over $60\%$ for the random-2 network and ASR of over $20\%$ for the Ring network.

In this paper, we aim to develop an adversarial robust DFL protocol that is capable of self-healing with built-in resilience. Our method is inspired by two pruning-based defense CLP (Zheng et al., 2022) and ANP (Wu & Wang, 2021), both built on the insight to detect the small number of poisoned parameters within the model and remove them afterward.

Technically, CLP utilizes the estimated Lipscheness constant to detect the poisoned neurons, while ANP identifies the poisoned neurons as those sensitive to artificial perturbation. However, both CLP and ANP are originally proposed for defense in centralized training, and they do not account for the fact that *the poisoned data structure in DFL is substantially different than that in a centralized setting*, which is exploitable in a defense design. Particularly, the key difference is that the centralized poisoning setting typically contains a single dataset with a low volume of backdoor data. In contrast, training data in DFL is dispersed to multiple small datasets, with only a small number of them being poisoned with a high volume of backdoor samples. The presence of this unique data structure inspires us to ask:

*Is there a more reliable indicator to identify the poisoned parameters in DFL?*

Our subsequent study theoretically reveals that the diagonal of empirical FI produced from the poisoned dataset of an adversary can be a reliable indicator of poisoned parameters, as pruning them can significantly increase the poisoned local loss. Empirically, we analyze the pattern of diagonal of empirical FI of each client's data and uncover an interesting invariant property at the temporal and client levels. This indicates that the poisoned parameters are unique and formed early during per-client training. Hence, progressive FI-based model pruning may enable self-healing. Our subsequent study of three naive pruning methods shows that removing parameters with TopK diagonal of empirical FI can indeed recover the model from backdoor behavior. Unfortunately, a straightforward approach to FI-based pruning tends to incur a large accuracy loss for deployed models, showing that a drastic reduction of ASR is achieved at the cost of performance degradation for classification of benign samples. We argue that *the use of diagonal of empirical FI for model pruning should be enhanced with performance-preserving strategies to minimize the accuracy loss.*

This motivates us to develop SILENCER, a two-stage model pruning scheme. The stage-1 FI-based pruning-aware training enables each client to only maintain the parameters important to its local training data in the training phase. As a result, the impact of the subsequent pruning of those parameters outside of a client-specific active subspace would be minimized. In stage-2, which is performed after federated training, each client identifies the poisoned/dummy parameters by deriving consensus from other clients. Those poisoned/dummy parameters are then removed them from its model. Results show that SILENCER lowers ASR by up-to 90.87% compared to DFedAvg without backdoor defense, while maintaining benign accuracy with minimal performance loss of up to 2.39%.

This paper makes three original contributions.

- We systematically study the risk of backdoor attacks in DFL with different P2P topologies.
- To the best of our knowledge, we are the first to identify and augment the diagonal of empirical Fisher information as an effective indicator for pruning poisoned parameters.
- We develop a two-stage method with pruning awareness to remedy the benign accuracy loss caused by immediate pruning. Experiments in various attack settings (poisoned ratios, attacker numbers) and DFL settings (data distribution and P2P topology) show the robustness of SILENCER.

## 2 RELATED WORK

**Decentralized Federated Learning.** DFL (Dai et al., 2022; Lalitha et al., 2018; Roy et al., 2019; Sun et al., 2022; Shi et al., 2023) is a decentralized extension of conventional federated learning (FL) without central coordination for aggregation (McMahan et al., 2016). See Appendix A.1 for details.

**Backdoor attack and defense.** Backdoor attack (Gu et al., 2017) is known to be detrimental to federated learning (Bagdasaryan et al., 2018). Several backdoor variants were proposed recently, e.g., edge-case backdoor (Wang et al., 2020), Neurotoxin (Zhang et al., 2022d), DBA (Xie et al., 2019) and stealthy model poisoning Bhagoji et al. (2019). Backdoor defenses are proposed for conventional federated learning, with three main categories: i) Certified robustness (Xie et al., 2021; Cao et al., 2022; Sun et al., 2021; Alfarra et al., 2022), which studies how to improve the robustness of model

with theoretical certification, ii) adversarial training (Zizzo et al., 2020; Shah et al., 2021; Zhang et al., 2023; 2022a), which studies how to improve model robustness by generating adversarial samples in training phase, iii) robust aggregation Blanchard et al. (2017); Ozdayi et al. (2021); Guerraoui et al. (2018); Cao et al. (2020); Pillutla et al. (2022); Andreina et al. (2021); Tolpegin et al. (2020); Chow et al. (2023); Li et al. (2020b); Guo et al. (2021); Zhang et al. (2022c); Rieger et al. (2022); Kumari et al. (2023), which studies how to identify the poisoned models in the training phase and exclude them from aggregation. A more detailed discussion on existing defense is provided in Appendix A.2.

**Pruning-aware training.** The core idea of pruning-aware training (Alvarez & Salzmann, 2017) is to maintain a sparse or nearly sparse model in training, so that the model being pruned after training still maintains comparable accuracy. Zimmer et al. (2022); Miao et al. (2021) utilize stochastic Frank-Wolfe algorithm to solve a k-sparse polytope constraint, leading to a large fraction of the parameters having a small magnitude throughout the training phase. Sparse training (Mocanu et al., 2018; Liu et al., 2021b; Evci et al., 2020; Liu et al., 2021a; Evci et al., 2022; Li et al., 2021a; 2020a) represents another category of work to achieve pruning-awareness, which, for example, enforces sparse binary masks on the model to de-activate some of the parameters throughout training.

To our best knowledge, this is the first systematic study of backdoor attack/defense in DFL (despite a concurrent work (Yar et al., 2023) focusing on peer-to-peer federated learning), and Silencer is also the first solution within the broad area of general backdoor defense that utilizes the diagonal of FI and two-stage pruning-aware training to boost its resilience against backdoor attack.

## 3 PRELIMINARIES

**Threat model.** Assume there are $M$ clients in the system, captured by a set $\mathcal{M}$, and assume a small subset of clients are adversaries, captured by a set $\mathcal{A}$. Each client has a local dataset $D_i$, but the attackers use another backdoor dataset $\tilde{D}_i$ for launching backdoor attack. Formally, the shared objective of adversaries is to manipulate other clients model $\boldsymbol{w}_j$ such that,

$$\min_{\boldsymbol{w}_j} \frac{1}{M} \left( \sum_{i \in \mathcal{A}} \tilde{f}_i(\boldsymbol{w}_j) + \sum_{i \in \mathcal{M}/\mathcal{A}} f_i(\boldsymbol{w}_j) \right) \tag{1}$$

where $\tilde{f}_i(\boldsymbol{w}_j) \triangleq \frac{1}{|\tilde{\mathcal{D}}_i|} \sum_{(\boldsymbol{x}_n y_n) \in \tilde{\mathcal{D}}_i} \mathcal{L}(\boldsymbol{w}_j; \boldsymbol{x}_n, y_n)$ is the malicious objective of attackers (to minimize the empirical loss of the backdoor dataset), $f_i(\boldsymbol{w}_j) \triangleq \frac{1}{|\mathcal{D}_i|} \sum_{(\boldsymbol{x}_n y_n) \in \mathcal{D}_i} \mathcal{L}(\boldsymbol{w}_j; \boldsymbol{x}_n, y_n)$ is the benign objective.

**Fisher information (FI).** Inspired by (Theis et al., 2018), we first reveal how the the local loss of a model changes according to a small perturbation $\epsilon$ on one specific coordinate, as follows.

$$f_i(\boldsymbol{w}_j + \epsilon \boldsymbol{e}_k) - f_i(\boldsymbol{w}_j) \approx \frac{1}{N_i} \sum_{n=1}^{N_i} \left( -g_k^{(n)} \epsilon + \frac{1}{2} H_{k,k}^{(n)} \epsilon^2 \right), \tag{2}$$

where $\boldsymbol{e}_k$ is a unit vector which is zero everywhere except at its $k$-th coordinate, $g_k^{(n)}$ is the coordinate-wise gradient over a data sample, $H_{k,k}^{(n)}$ is the diagonal of the Hessian on $k$-th position, and $N_i$ is the number of samples within a local dataset. As shown, the local loss increases more drastically when perturbing the coordinate $k$ with larger $H_{k,k}^{(n)}$. That means that the diagonal of Hessian measures the importance of a parameter in terms of maintaining a small loss over a sample. However, computing Hessian is expensive. We therefore use the diagonal of FI as an approximation, as follows:

$$H_{kk} \approx \mathbb{E}_{P(y|\boldsymbol{x})} \left[ \left( \frac{\partial}{\partial \boldsymbol{w}_k} \log Q_{\boldsymbol{w}}(y \mid \boldsymbol{x}) \right)^2 \right], \tag{3}$$

where $P(y|\boldsymbol{x})$ is the data distribution, and $Q_w(y|\boldsymbol{x})$ is the output distribution of a model. If $P$ and $Q_w$ are equal (i.e., the model is "well-trained") and the model is twice differentiable, the approximation becomes exact. Finally, replacing Hessian, we use the following metric (diagonal of empirical FI) to empirically measure the importance of coordinates given a model $\boldsymbol{w}$ and over a local dataset $\mathcal{D}_i$:

$$F_i(\boldsymbol{w}) = \frac{1}{N_i} \sum_{n=1}^{N_i} \left( \nabla_{\boldsymbol{w}} \log Q_{\boldsymbol{w}}(y_n \mid \boldsymbol{x}_n) \right)^2, \tag{4}$$

The metric is useful in determining the useful parameters for a potentially poisoned dataset $\mathcal{D}_i$ and thereby helping us to identify the poisoned parameters (i.e., useful parameters on a poisoned dataset).

# 4 METHODOLOGY

## 4.1 FI INVARAINCE PHENOMENON AND FISHER-GUIDED PRUNING

We first verify that the diagonal of empirical FI (See Eq. (4) can be used to approximate that of the Hessian in Figure 2a. Given that the diagonal of empirical FI reveals the importance of model parameters over the local dataset, a question arises: *Can we identify the poisoned parameters based on the diagonal of their empirical FI?* To answer this, we visualize the pattern in Figure 2b.

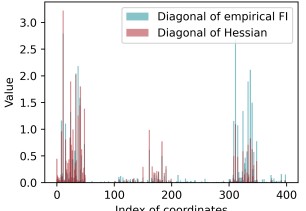

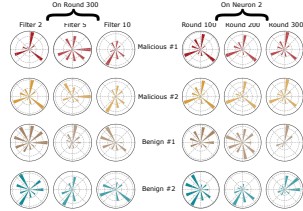

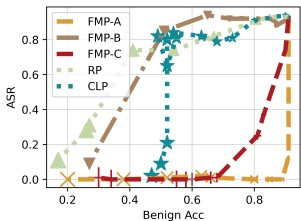

(a) Statistical comparison between diagonal of Hessian and that of empirical FI over the last layer of models. Empirical FI approximates Hessian well in coordinate level.

(b) Visualization of diagonal of empirical FI over first layer of the model. Each bar represents the FI on one coordinate and each disk represents a filter. The poisoning pattern exhibits *temporal/client-level invariance*.

(c) Pruning on a poisoned model. RP refers to random pruning, which randomly prune the network. CLP is from (Zheng et al., 2022) . Each point represents the ASR and BA under one specific pruning rate reflected by its size.

Figure 2: Study of Fisher-guided pruning on FashionMnist. (a) is run with a simple two-layer MLP model (because obtaining hessian requires massive computation). (b)(c) are run with a CNN model).

We uncover an interesting phenomenon on FI invariance property over different clients and rounds. [1]

- **Client-level invariance**. On the left of Figure 2b, the patterns of FI between two malicious clients are invariant. In contrast, the FI-pattern between two benign clients are highly discrepant. This indicates that the attackers share the same group of parameters to maintain their low poisoning loss. Next, the FI pattern between the attackers and the benign clients are quite different, indicating that the poisoned parameters are unique to the attackers and are not used by the benign clients.
- **Temporal-level invariance**. On the right of Figure 2b, we visualize the FI over checkpoints of different rounds. We observe that each attacker shares the same FI pattern across the training rounds, which indicates that it use specific parameters for poisoning at the early stage, and such poisoned parameters remain unchanged in subsequent rounds. This phenomenon is not observable for benign clients– their FI pattern are continuously evolving across rounds of federated learning.

Given that different adversaries share the unique and yet temporally invariant FI patterns of their poisoned parameters, an intuitive solution is to clean the poisoned model by pruning those poisoned parameters and recover it from poisoning. Here are three baseline FI-based pruning methods:

- **Fisher Information Pruning on Adversary (FIP-A)**. We ask one malicious client to report their empirical FI calculated from their poisoned dataset. Subsequently, we prune the TopK parameters with the highest diagonal of empirical FI within the well-trained global model
- **Fisher Information Pruning on Benign clients (FIP-B)**. We ask one benign client to report their empirical FI calculated from their poisoned dataset and prune the topK of them like FIP-A.
- **Fisher Information Pruning on Combined clients (FIP-C)**. We ask every clients ($M$ of them) in the system to report their empirical FI. We prune the model $M$ times, and each time we prune the TopK parameters with the highest diagonal of empirical FI for each client.

Figure 2c shows the measurement results. FIP-A achieves the best defense performance – its ASR is significantly reduced with a small pruning rate. FIP-C demonstrates comparable defense performance. FIP-B performs worse than RP when using a small pruning rate. The results demonstrate that the diagonal of empirical FI of a malicious client can reliably reveal the poisoned parameters, and using FI invariant property to identify and remove them can effectively correct/minimize the poisoning behavior of the model. However, given that FIP-C assumes no knowledge of the malicious clients, the pruning could lead to a non-negligible downgrade of benign accuracy ($\approx 20\%$ loss) when the defense aims to achieve near zero ASR. This motivates us to propose the *pruning-aware defense*. By pruning awareness, we imply that in the training phase, the clients should train on sparse models, such that the effect of pruning after training can be mitigated.

---

[1] See checkpoints in `https://www.dropbox.com/sh/ibqi2rjnxrn2p8n/AACCcEc-SA4ruMxjvXPgzLC_a?dl=0`.

### 4.2 AUTO-HEALING WITH DYNAMIC TWO-STAGE PRUNING-AWARE TRAINING

Figure 3 shows the two-stage pruning scheme named SILENCER. In stage-1, we achieve pruning awareness by asking each client to train only a subset of parameters within the model, and this subset of parameters should be those that are deemed important per each client's local dataset. In stage-2, we ask the clients to prune out those parameters that are deemed unimportant by most of the clients (i.e., those poisoned/dummy parameters). Because the local training is conducted on a sparse model with most poisoned/dummy parameters removed, the performance degradation caused by stage-2 can be controlled, i.e., it achieves pruning awareness.

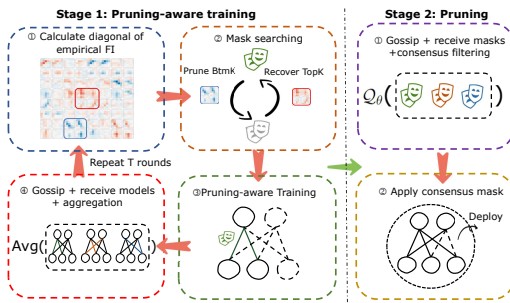

Figure 3: Procedural illustration for Silencer.

**Mask initialization.** At round 0, we enforce the same binary mask $\boldsymbol{m}_{i,0}$ with overall sparsity $s$ to initialize masks for every participating client. To perform pruning-aware training, each client selects the subset of the parameters based on the mask it has and only trains the parameters for the coordinates that are 1 in the mask, while keeping other coordinates untouched. For maintaining practical performance, we follow (Evci et al., 2020) to use ERK for mask initialization, in which different layers in the model are designated different sparsity. (See Appendix C.2 for the detailed initialization technique).

**Mask searching.** Given that the FI pattern of a benign client changes across rounds (recall Section 4.1), we propose to dynamically evolve the mask of each client for round $t + 1$ based on the diagonal of empirical FI obtained from the current model. Specifically. we adopt an

---

**Algorithm 1** Silencer (run by the $i$-th client)

**input** Iteration $T$; Local steps $K$; Learning rate $\eta$; Pruning rate $\alpha_t$; Initial model $\boldsymbol{w}_{i,0}$; Neighbours, $\mathcal{N}_{i,t}$.
**output** Models for deployment $\tilde{\boldsymbol{w}}_{i,T}$
1: Initialize mask $\boldsymbol{m}_{i,0}$ with overall sparsity $s$.
2: **for** $t = 0, 1, \ldots, T - 1$ **do**
3:      $\boldsymbol{w}_{i,t,0} = \boldsymbol{w}_{i,t} \odot \boldsymbol{m}_{i,t}$
4:      Compute $F_i(\boldsymbol{w}_{i,t,0})$) per Eq. (4)
5:      $\boldsymbol{m}_{i,t+\frac{1}{2}} = \boldsymbol{m}_{i,t} - \text{ArgBtmK}_{\alpha_t}(F_i(\boldsymbol{w}_{i,t,0}))$
6:      $\boldsymbol{m}_{i,t+1} = \boldsymbol{m}_{i,t+\frac{1}{2}} + \text{ArgTopK}_{\alpha_t}(F_i(\boldsymbol{w}_{i,t,0}))$
7:      **for** $k = 0, 1, \ldots, K - 1$ **do**
8:          $\boldsymbol{w}_{i,t,k+1} = \boldsymbol{w}_{i,t,k} - \eta \boldsymbol{m}_{i,t+1} \odot \nabla f_i(\boldsymbol{w}_{i,t,k}; \xi)$
9:      **end for**
10:      $\boldsymbol{w}_{i,t+\frac{1}{2}} = \boldsymbol{w}_{i,t} - \boldsymbol{m}_{i,t+1} \odot (\boldsymbol{w}_{i,t,0} - \boldsymbol{w}_{i,t,K})$
11:      Gossip $\boldsymbol{w}_{i,t+\frac{1}{2}}$ to her neighbours $\mathcal{N}_{i,t}$.
12:      $\boldsymbol{w}_{i,t+1} = \text{Avg}(\{\boldsymbol{w}_{i',t+\frac{1}{2}}\}_{i' \in \mathcal{N}_i})$
13: **end for**
14: Gossip masks and obtain $\{\boldsymbol{m}_{1,T}, \ldots, \boldsymbol{m}_{M,T}\}$
15: Deploy $\tilde{\boldsymbol{w}}_{i,T} = \mathcal{Q}_\theta(\boldsymbol{m}_{1,T}, \ldots, \boldsymbol{m}_{M,T}) \odot w_{i,T}$

---

alternative mask pruning/mask reclamation method before each local training iteration, as follows.

1. **Mask pruning.** We first prune the parameters with the BottomK diagonal of empirical FI from the subset of the active (selected) parameter, which removes the less important parameters over the client's local dataset. Formally, this process can be formulated as follows:

$$\boldsymbol{m}_{i,t+\frac{1}{2}} = \boldsymbol{m}_{i,t} - \text{ArgBtmK}_{\alpha_t}(F_i(\boldsymbol{w}_{i,t,0})), \tag{5}$$

where $\text{ArgBtmK}_{\alpha_t}(F_i(\boldsymbol{w}_{i,t,0}))$ returns the $\alpha_t$ percentage of smallest coordinates of diagonal of empirical FI and mask them to 1, indicating that they will be pruned. We use cosine annealing to decay $\alpha_t$ from the initial pruning rate $\alpha_0$. The decay process is postponed to the Appendix C.2.

2. **Mask reclamation.** This step is performed after mask pruning, aiming to recover the same amount of parameters into the active subset. One approach is to choose the coordinates with TopK diagonal of empirical FI to recover, as follows:

$$\boldsymbol{m}_{i,t+1} = \boldsymbol{m}_{i,t+\frac{1}{2}} + \text{ArgTopK}_{\alpha_t}(F_i(\boldsymbol{w}_{i,t,0}) \tag{6}$$

where $\text{ArgTopK}_{\alpha_{t-1}}(F_i(\boldsymbol{w}_{i,t,0}))$ returns the $\alpha_t$ percentage of the largest coordinates of diagonal of empirical FI and mask them to 1, indicating that they will be recovered.

**Pruning-aware training.** Upon the completion of the first two tasks in a given round, each participating client performs multiple local steps to train a sparse surrogate model with a subset of parameters enforced by the mask. Specifically, for local step $k = 0, \ldots, K - 1$,

$$\boldsymbol{w}_{i,t,k+1} = \boldsymbol{w}_{i,t,k} - \eta \boldsymbol{m}_{i,t+1} \odot \nabla f_i(\boldsymbol{w}_{i,t,k}; \xi), \tag{7}$$

where $\xi$ is a piece of random sample within the local dataset, $\eta$ is the learning rate, and $\odot$ denotes Hadamard product. Intuitively, the binary mask $\boldsymbol{m}_{i,t+1}$ is applied to the gradient in every local

step, which ensures that along the training process, the model is sparse everywhere except for those coordinates within the active subset. By this means, if the poisoned parameters are outside of the benign client's active subset, the benign clients are training a sparse model without malicious parameters involved, i.e., they are aware of the pruning after training.

**Decentralized Aggregation.** For each round of federated learning, upon completion of local pruning-aware training, each client will first update its dense model using the gradient update of the surrogate model obtained from the local model training phase. Then the client will send the updated dense model to its neighbors according to the P2P topology, and it will also receive from the neighbor peers their local trained models. After collecting all the models, an average aggregation is performed, which yields the dense model for the next round. Assume $\mathcal{N}_i$ is the neighbour set, which also includes the sender herself for simplicity. This process can be formulated as follows:

$$\boldsymbol{w}_{i,t+1} = \text{Avg}(\{\boldsymbol{w}_{i,t} - \boldsymbol{m}_{i,t+1} \odot (\boldsymbol{w}_{i,t,0} - \boldsymbol{w}_{i,t,K})\}_{i \in \mathcal{N}_i}) \qquad (8)$$

**Consensus filtering**. After the iterative training has reached the total number of rounds, pre-defined at federated learning initialization time, SLIENCER will perform consensus based pruning to remove the poisoned/dummy parameters. Since the mask of each client reflects those parameters that they deem important, we propose to utilize a consensus filtering (CF) method to remove those parameters that have relatively less appearance in the active (selected) subsets of parameters by all the clients.

$$[\mathcal{Q}_\theta(\boldsymbol{m}_{1,T}, \ldots, \boldsymbol{m}_{M,T})]_j = \begin{cases} 1 & \sum_{i=1}^{M}[\boldsymbol{m}_{i,T}]_j \geq \theta \\ 0 & \text{Otherwise} \end{cases}, \qquad (9)$$

where $\theta$ is the threshold for CF, and $[\cdot]_j$ indexes the $j$-th coordinate of a vector. By applying the consensus mask produced by CF into each client's dense model, i.e., $\boldsymbol{w}_{i,T} \odot \mathcal{Q}_\theta(\boldsymbol{m}_{1,T}, \ldots, \boldsymbol{m}_{M,T})$, those parameters that have appearances smaller than $\theta$ are sparsified to 0. Our high-level idea is that the poisoned parameters would only be deemed important for adversaries (who account for minority), therefore those parameters that are shared by minority will most likely be the poisoned ones.

## 5 EXPERIMENTS

### 5.1 EXPERIMENT SETUP

**Datasets and models.** Five datasets are used: FashionMnist, GTSRB, CIFAR10/100, and TinyImagenet. We use a CNN model for FMnist and a ResNet9 for others. See Appendix D.1.

**Attack methods.** We simulate four representative backdoor attacks: BadNet (Gu et al., 2017), DBA (Xie et al., 2019), Sinusoidal (Barni et al., 2019), Neurotoxin (Zhang et al., 2022d), and design two adaptive attacks, which assumes the knowledge of our defense and attempts to break it.

**Defense Baselines.** We use vanilla D-FedAvg (Sun et al., 2022) (without defense) as a baseline, and compare SILENCER with three SOTA defenses: D-RLR (Ozdayi et al., 2021), D-Krum (Blanchard et al., 2017), D-FLTrust (Yin et al., 2018), D-Bulyan (Guerraoui et al., 2018). All baselines are defenses originally for conventional FL, and we adapt them to the DFL settings. See Appendix B.1.

**Evaluation metrics**. Three metrics are used to evaluate the defense performance:

- **Benign Acc.** Benign accuracy measures the Top-1 accuracy of a model over benign samples.
- **ASR.** Attack Success Ratio (ASR) measures the ratio of backdoor samples with triggers to be classified to the target label. The lower this metric is, the better the performance of the defense is.
- **Backdoor Acc.** Backdoor Acc measures the Top-1 accuracy of the model under backdoor attack.

In DFL, each client owns and deploys its unique model. Therefore, all three metrics are measured using the average statistics of all models of participating clients within the system.

**DFL Simulation setting.** We simulate $M = 40$ clients, for each of which we distribute a subset of data in either an IID or a Non-IID way. For Non-IID, we use Dirichlet distribution (Hsu et al., 2019) for simulation, with its default parameter set to 0.5. In the backdoor simulation, we simulate $N$ out of $M$ clients to be adversaries, each with a poisoned dataset in the ratio of $p$. $N = 4$ and $p = 0.5$ are the default settings. We also simulate three different types of DFL topology, as reflected by the neighbor set $\mathcal{N}_{i,t}$. Particularly, *Ring* topology involve $\{i-1, i, i+1\}$ as its neighbours set $\mathcal{N}_{i,t}$ in every round, *random* topology uses a random $\mathcal{N}_{i,t}$ for each round, and *full* topology involve all clients in $\mathcal{N}_{i,t}$.

**Hyper-parameters.** For every defense, the learning rate and weight decay used in the local optimizer are fixed to $0.1$ and $10^{-4}$, and the number of local epochs and batch size are set to 2 and 64 respectively. For SILENCER, we set the overall sparsity to $s = 0.25$, the consensus threshold to $\theta = 20$, and the initial pruning rate to $\alpha_0 = 10^{-4}$. The number of training rounds is fixed to 300. See Appendix 5.1 for detailed setup.

## 5.2 MAIN EVALUATION

In our main evaluation, we use CIFAR10 as the default dataset, BadNet as the default attack, and Random-8 as the default topology, unless otherwise specified. Bold number in the table corresponds the best performance among all the *defenses*.

**Convergence.** Figure 4 shows the convergence w.r.t communication rounds. SILENCER consistently outperforms the SOTA defenses with low ASR (<10%) and high benign accuracy.

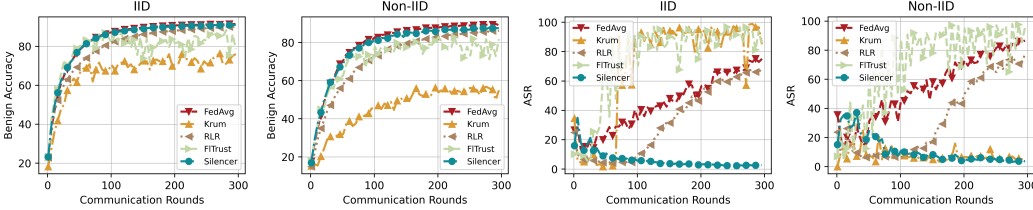

Figure 4: Convergence and defense performance under different defenses.

**Defense efficacy on poison ratio.** We show in Table 1 the defense performance under different poison ratios. Compared to FedAvg without defense, Silencer achieves consistently low ASR (<11%), a minor drop of benign accuracy (<2% of reduction), and significant enhancement of backdoor accuracy (>10%). Another observation is that the defense performance is better when the poison ratio is higher. This can be explained by the fact that the poisoned parameters will be more outstanding (in terms of empirical FI) when the backdoor data constitute a larger portion of the poisoned dataset.

Table 1: Defense efficacy with varying poison ratio $p$ under CIFAR10.

| Methods | Benign Acc (%) ↑ | | | | | ASR (%) ↓ | | | | | Backdoor Acc (%) ↑ | | | | |
|---|---|---|---|---|---|---|---|---|---|---|---|---|---|---|---|
| (IID) | clean | p=.05 | p=.2 | p=.5 | p=.8 | clean | p=.05 | p=.2 | p=.5 | p=.8 | clean | p=.05 | p=.2 | p=.5 | p=.8 |
| D-FedAvg | 91.92 | 91.92 | 92.05 | 91.42 | 91.54 | 1.61 | 14.44 | 33.12 | 75.09 | 98.47 | 89.50 | 78.43 | 64.08 | 23.66 | 1.46 |
| D-RLR | **90.41** | 90.54 | 90.14 | 90.02 | 90.03 | 1.69 | 2.26 | 5.41 | 68.78 | 94.08 | **87.77** | 86.61 | 84.16 | 30.96 | 5.98 |
| D-Krum | 79.40 | 81.39 | 82.05 | 76.04 | 70.58 | 3.71 | 3.29 | 4.48 | 85.00 | 99.91 | 77.03 | 77.68 | 78.17 | 2.32 | 0.04 |
| D-FLTrust | 87.85 | 88.02 | 86.41 | 78.64 | 81.89 | **1.62** | 20.09 | 40.94 | 83.77 | 58.52 | 84.56 | 69.60 | 48.76 | 11.37 | 28.23 |
| D-Bulyan | 90.17 | 90.14 | 90.08 | 88.66 | 86.78 | 1.76 | 2.14 | 55.66 | 96.79 | 99.30 | 87.62 | 86.33 | 40.53 | 2.46 | 0.49 |
| Silencer | 90.14 | **90.73** | **90.68** | **91.04** | **90.33** | 2.08 | **1.89** | **3.85** | **2.67** | **2.92** | 87.51 | **87.68** | **86.27** | **87.22** | **86.56** |

| Methods | Benign Acc (%) ↑ | | | | | ASR (%) ↓ | | | | | Backdoor Acc (%) ↑ | | | | |
|---|---|---|---|---|---|---|---|---|---|---|---|---|---|---|---|
| (Non-IID) | clean | p=.05 | p=.2 | p=.5 | p=.8 | clean | p=.05 | p=.2 | p=.5 | p=.8 | clean | p=.05 | p=.2 | p=.5 | p=.8 |
| D-FedAvg | 88.98 | 89.38 | 89.28 | 89.08 | 88.77 | 1.83 | 30.06 | 64.02 | 86.58 | 96.98 | 86.41 | 66.59 | 33.96 | 12.52 | 3.17 |
| D-RLR | 85.91 | **87.32** | 86.86 | 86.07 | 86.52 | 2.58 | 12.26 | 31.21 | 77.55 | 85.45 | 82.76 | 75.46 | 62.38 | 20.09 | 14.74 |
| D-Krum | 48.44 | 55.14 | 56.12 | 54.95 | 50.08 | 4.57 | 7.16 | 4.80 | 5.72 | 99.29 | 45.26 | 53.03 | 54.93 | 51.63 | 0.01 |
| D-FLTrust | 74.14 | 82.89 | 78.60 | 81.74 | 82.71 | 2.64 | 22.35 | 36.85 | 95.96 | 31.41 | 80.94 | 63.34 | 51.36 | 7.84 | 79.31 |
| D-Bulyan | 80.42 | 80.34 | 80.82 | 79.21 | 76.25 | 3.98 | **3.72** | **3.42** | 96.73 | 99.39 | 75.09 | 73.84 | 74.18 | 1.34 | 0.29 |
| Silencer | **87.35** | 87.31 | **87.69** | **87.48** | **87.30** | **1.14** | 10.96 | 6.95 | **3.63** | **3.26** | **84.42** | **75.62** | **78.06** | **82.01** | **81.74** |

**Defense efficacy on varying attacker ratio.** We fix the poison ratio to 0.5 and vary the attackers ratio to $\{0, 0.1, 0.2, 0.3, 0.4\}$. The results are shown in Figure 5. As shown, Lockdown *consistently* achieves low ASR (at minimum 30% ASR reduction compared to FedAvg when attackers ratio is 0.4), and high benign accuracy in all groups of experiments (at maximum 5% drop of benign accuracy compared to FedAvg). In contrast, RLR and Krum are *sensitive* to attacker ratio and *fail* in defense when attacker ratio is 0.4 (no ASR reduction for Krum, and at maximum 10% reduction for RLR).

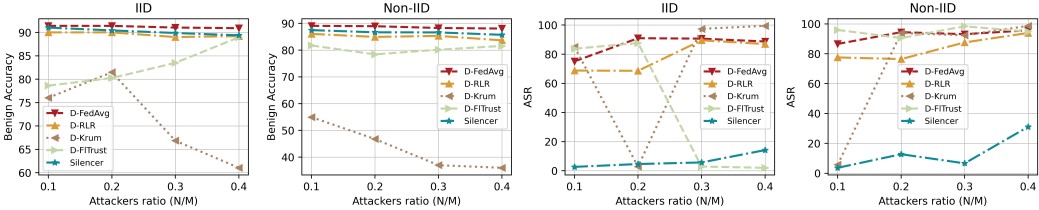

Figure 5: Defense performance under different adversary number.

**Defense efficacy on varying topology.** In Table 2, we show that Silencer can be generalized to different topologies. For each topology, ASR is reduced to $< 6\%$ in either IID and Non-IID setting, while only incuring $< 2\%$ benign accuracy loss in most settings.

Table 2: Performance on varying topology.

| | Ring | | Random-2 | | Random-4 | | Random-8 | | Full | |
|---|---|---|---|---|---|---|---|---|---|---|
| Methods (IID) | BA ↑ | ASR ↓ | BA ↑ | ASR ↓ | BA ↑ | ASR ↓ | BA ↑ | ASR ↓ | BA ↑ | ASR ↓ |
| D-FedAvg | 86.50 | 23.48 | 90.91 | 65.07 | 91.50 | 71.98 | 91.42 | 75.09 | 91.91 | 89.02 |
| D-RLR | 16.97 | 7.56 | 12.80 | 21.46 | 91.48 | 71.91 | 90.02 | 68.78 | 81.26 | 4.53 |
| D-Krum | 55.73 | 5.62 | 82.36 | 41.33 | 80.52 | 97.26 | 76.04 | 85.00 | 77.46 | 5.21 |
| D-FLTrust | 68.89 | 25.57 | 90.30 | 88.97 | 91.18 | 94.43 | 78.64 | 83.77 | 84.25 | 85.61 |
| D-Bulyan | 81.41 | 17.65 | 91.01 | 66.99 | 91.00 | 96.85 | 88.66 | 96.79 | 8.88 | 51.33 |
| Silencer | **84.83** | **3.15** | 89.98 | **2.83** | 90.71 | **2.59** | **91.04** | **2.67** | 90.94 | **2.08** |
| Methods (Non-IID) | BA ↑ | ASR ↓ | BA ↑ | ASR ↓ | BA ↑ | ASR ↓ | BA ↑ | ASR ↓ | BA ↑ | ASR ↓ |
| D-FedAvg | 74.42 | 30.77 | 86.03 | 74.47 | 88.49 | 79.38 | 89.08 | 86.58 | 90.48 | 89.44 |
| D-RLR | 14.21 | 10.74 | 12.24 | 8.34 | 87.46 | 52.22 | 86.07 | 77.55 | 69.96 | 11.17 |
| D-Krum | 31.71 | 8.81 | 57.64 | 30.57 | 55.49 | 7.52 | 54.95 | 5.72 | 38.66 | 6.58 |
| D-FLTrust | 47.20 | 13.82 | 79.88 | 67.39 | 84.26 | 88.13 | 81.74 | 95.96 | 63.06 | 70.05 |
| D-Bulyan | 57.27 | 13.98 | **86.99** | 82.96 | 85.55 | 97.19 | 79.21 | 96.73 | 12.51 | **0.00** |
| Silencer | **69.74** | **5.91** | 86.23 | **3.57** | **88.02** | **3.99** | **87.48** | **3.63** | 87.41 | 3.54 |

**Defense against adaptive attack.** In a decentralized environment, it is almost impossible to force every client (including the attackers) to run the designated protocol. We simulate three adaptive attacks dubbed FixMask, BenignSearch and Curiosity to show whether the attacker can evade Silencer by not obeying the designated mask searching process (see description in Appendix C.3). Our results in Table 3 show that under proper $\alpha_0$, FixMask and BenignSearch cannot evade Silencer defense. However, we find that Curiosity attacks can always evade Silencer's defense. In Curiosity , the adversaries infer the consensus mask, and project updates into it, giving the chance of poisoned parameters to evade consensus filtering. But we need to note that some conditions need be be made to achieve the attack (See Appendix C.3).

Table 3: Silencer on adaptive attack.

| Methods (IID) | Benign Acc(%) ↑ | ASR(%) ↓ | Backdoor Acc(%) ↑ |
|---|---|---|---|
| BadNet | 91.04 | 2.67 | 87.22 |
| FixMask ($\alpha_0 = 0.0001$) | 90.27 | 32.27 | 62.77 |
| FixMask ($\alpha_0 = 0.1$) | 90.86 | 2.77 | 87.67 |
| BenignSearch ($\alpha_0 = 0.0001$) | 91.21 | 65.24 | 35.02 |
| BenignSearch ($\alpha_0 = 0.1$) | 91.13 | 1.83 | 88.69 |
| Curiosity | 91.16 | 58.53 | 41.38 |
| Methods (Non-IID) | Benign Acc(%) ↑ | ASR (%) ↓ | Backdoor Acc(%) ↑ |
| BadNet | 87.48 | 3.63 | 82.01 |
| FixMask ($\alpha_0 = 0.0001$) | 87.34 | 81.43 | 18.71 |
| FixMask ($\alpha_0 = 0.1$) | 87.68 | 2.10 | 85.34 |
| BenignSearch ($\alpha_0 = 0.0001$) | 87.49 | 68.99 | 38.98 |
| BenignSearch ($\alpha_0 = 0.1$) | 87.70 | 9.67 | 82.08 |
| Curiosity | 88.27 | 91.03 | 7.16 |

**Communication and inference complexity.** We study the communication/inference complexity of Silencer and its communication-reduction variant (Silencer-CR, see Appendix C.4) in Table 4. Our results show that under overall sparsity $s = 0.75$, Silencer can reduce 0.75x model inference complexity, while Silencer-CR enjoys 0.75x communication reduction in addition to the inference complexity reduction. Our results also indicate that Silencer-CR would not degrade the defense performance (in terms of ASR and benign acc) without the presence of an adaptive attack.

Table 4: Communication and # of parameters for Silencer under default setting (IID). The communication overhead is the sum of that of all clients in each round.

| Methods | Benign Acc(%) ↑ | ASR ↓ | Comm Overhead ↓ | # of params ↓ |
|---|---|---|---|---|
| D-FedAvg | 91.42 | 75.09 | 2.10GB (1x) | 6.57M (1x) |
| Silencer | 91.04 | 2.67 | 2.10GB (1x) | 1.643M (0.25x) |
| Silencer-CR | **91.28** | **2.12** | **0.525GB** (0.25x) | **1.643M** (0.25x) |

**Robustness to Non-IID.** We show the robustness of Silencer to Non-IID extent in Table 5. Silencer reduces ASR to smaller than 11% in all the settings, and controls the BA loss within 2.78%. Another

interesting observation is that Silencer seems to reduce more benign accuracy when the non-iid level increases. This may be because the mask obtained from the clients are more heterogeneous in this case, which results in a weaker consistency mask in the mask searching process, and thereby leading to performance degradation after consensus filtering stage.

Table 5: Performance of Silencer on Non-IID extent. Blue number is the baseline without defense.

| $\alpha$ | Benign Acc(%) $\uparrow$ | ASR(%) $\downarrow$ | Backdoor Acc(%) $\uparrow$ |
|---|---|---|---|
| 0.1 | 76.64 (78.33) | 5.88 (64.80) | 71.77 (34.38) |
| 0.3 | 85.09 (87.87) | 10.17 (90.05) | 74.36 (19.92) |
| 0.5 | 87.48 (89.08) | 3.63 (86.58) | 82.01 (12.52) |
| 0.7 | 88.29 (89.69) | 2.93 (76.30) | 85.07 (27.67) |
| 1 | 89.58 (90.81) | 3.35 (75.18) | 85.52 (26.62) |
| IID | 91.04 (91.42) | 2.67 (75.09) | 87.22 (23.66) |

**Generalization to varying datasets.** We show our evaluation results on FashionMnist, GTSRB, CIFAR10/100, and TinyImagenet in Table 6. As shown, Silencer achieves SOTA defense efficacy (compared with D-FedAvg without defense, with up-to 89.19%, 91.48%, 86.74%, 78.5%, and 96.39% reduction of ASR on the five datasets. With pruning awareness, the loss of benign accuracy is also mitigated, with up-to 1.68%, 2.99%, 1.94% and 3.68% drop compared to D-FedAvg.

Table 6: Performance on varying datasets.

| / | FashionMnist | | GTSRB | | CIFAR10 | | CIFAR100 | | TinyImagenet | |
|---|---|---|---|---|---|---|---|---|---|---|
| Methods (IID) | BA $\uparrow$ | ASR $\downarrow$ | BA $\uparrow$ | ASR $\downarrow$ | BA $\uparrow$ | ASR $\downarrow$ | BA $\uparrow$ | ASR $\downarrow$ | BA $\uparrow$ | ASR $\downarrow$ |
| D-FedAvg | 91.04 | 97.44 | 98.01 | 96.03 | 91.51 | 82.90 | 69.45 | 79.72 | 47.91 | 96.58 |
| D-RLR | **90.95** | 72.24 | **98.26** | 29.10 | 90.18 | 66.43 | **67.61** | 43.43 | **45.23** | 68.47 |
| D-Krum | 87.86 | **0.06** | 96.28 | 0.04 | 68.78 | 98.61 | 40.54 | 99.01 | 37.49 | 0.40 |
| D-FLTrust | 88.89 | 0.33 | 97.25 | **0.08** | 80.32 | 92.54 | 52.42 | 63.15 | 0.50 | **0.00** |
| Silencer | 89.36 | 4.06 | 98.17 | 0.59 | **90.52** | **2.89** | 66.29 | **1.22** | 45.13 | 0.52 |
| Methods (Non-IID) | BA $\uparrow$ | ASR $\downarrow$ | BA $\uparrow$ | ASR $\downarrow$ | BA $\uparrow$ | ASR $\downarrow$ | BA $\uparrow$ | ASR $\downarrow$ | BA $\uparrow$ | ASR $\downarrow$ |
| D-FedAvg | 89.87 | 94.80 | 98.31 | 94.33 | 88.94 | 91.57 | 68.29 | 72.73 | 47.99 | 96.72 |
| D-RLR | 89.59 | 66.01 | **98.63** | 43.78 | 86.71 | 64.97 | 62.93 | 42.75 | **46.54** | 75.33 |
| D-Krum | 80.13 | **0.25** | 93.30 | **0.04** | 52.28 | 5.32 | 30.75 | 97.37 | 31.90 | 0.57 |
| D-FLTrust | 82.30 | 0.69 | 96.14 | 1.08 | 79.41 | 93.58 | 52.86 | 61.01 | 7.71 | 14.19 |
| Silencer | **88.44** | 5.61 | 95.32 | 2.85 | **87.00** | **4.83** | **65.50** | **0.47** | 44.31 | **0.33** |

## 5.3 ABLATION AND HYPER-PARAMETER SENSITIVITY ANALYSIS

Our main conclusions are that i) ERK initialization can significantly mitigate the degradation of accuracy of the sparse model, and slightly reduce ASR. ii) Fisher information is important in the mask searching process, especially in the reclamation process. iii) Consensus filtering, which is responsible for pruning out the malicious parameters in the model, is also important. Another notable observation is that after consensus filtering, the model's benign accuracy can also be enhanced. Our hyper-parameter sensitivity experiment demonstrates that i) higher sparsity can gain better defense performance while zero sparsity actually reduces to D-FedAvg with no defense. ii) A sufficiently large exploration ratio ($\alpha_0$) is necessary for Silencer to work in practice. iii) the consensus threshold should be set depending on potential attacker numbers. With a larger number of attackers presented, the corresponding threshold should also be set larger. See Appendix D.3 for the missing results.

## 6 CONCLUSION

In this paper, we study the diagonal of empirical FI as an indicator to identify and remove the poisoned parameters. The usage is supported by theory and is systematically examined by empirical findings. In order to control the benign accuracy loss due to unawareness of pruning, we propose a two-stage pruning-aware defense dubbed Silencer. Empirical evidence shows that Silencer significantly reduces the risk of malicious backdoor attacks without sacrificing much on benign accuracy. Future works include studying how to generalize Silencer to other FL settings, e.g., personalized FL (Fallah et al., 2020; Deng et al., 2020; T Dinh et al., 2020), FL with distillation (Zhang et al., 2022b), etc.

## 7 REPRODUCIBILITY STATEMENT

We make the following effort to enhance the reproducibility of our results.

- For Silencer implementation, a link to an anonymous downloadable source is included in our abstract. We also briefly introduce the implementation of Silencer in Appendix C.2.

- We show a brief description of defense and attack baselines in Appendix B. The hyper-parameters and simulation settings are given in Section 5.1. Detailed setting and the hyper-parameter selection logistic can be found in Appendix D.1.

- For our motivation study, we provide a link to an anonymous Dropbox repository on page 4. The repository stores several poisoned model checkpoints trained by FedAvg without defense, which are utilized to investigate the pruning algorithm performance. We also provide the code of the proposed pruning algorithms within the same Dropbox repository. We hope this can help the community to better understand the FI invariance phenomenon.

- In Section 3, we clearly describe the threat model, and we also provide some theoretical results to motivate the use of empirical Fisher information as an indicator of the importance of each coordinate of parameters. This result is adapted and modified from (Theis et al., 2018), but should be self-contained in this paper's context.

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

**Organization of Appendix**

# A    MORE DISCUSSION ON EXISTING LITERATURE

We in this section provide more discussion on existing literature on DFL and backdoor defense in FL.

## A.1    ON DECENTRALIZED FEDERATED LEARNING

Decentralized federated learning (aka gossip learning) follows a P2P framework for model training. In DFL, federated aggregation is performed with decentralized coordination using a peer-to-peer communication protocol. There is no pre-defined federated server(s), which largely alleviates the communication bottleneck of the centralized server in conventional FL. In summary, DFL excels conventional FL in two main aspects.

- **Bandwidth bottleneck.** Centralized entity (the server) has been recognized as the bandwidth bottleneck in a classic server/client model, as the model uploading/distributing process heavily relies on the server. Through eliminating the need of a centralized server, and embracing a P2P communication model, the bandwidth bottleneck can be well resolved.

- **Monopolization and privacy concern.** DFL, as a P2P model in nature, mitigates monopolization and privacy concerns because the trained model is not exclusively owned by a centralized power. It can be challenging for a centralized power to misuse the model to make profits or try to extract private information from the model.

We present the vanilla version of D-FedAvg in Algorithm 2. In D-FedAvg, every client can be viewed as a server, who aggregates the post-trained model sent by her neighbors. Each client would exclusively have her unique version of the deployed model, i.e., $\boldsymbol{w}_{i,T}$ are different for different clients. This model absorbs knowledge from the clients it communicated before within $T$ rounds of training.

---

**Algorithm 2** D-FedAvg (adapted from Sun et al. (2022))

**input** Iteration $T$; Local steps $K$; Learning rate $\eta$; Initial model $\boldsymbol{w}_{i,0}$; Neighbours, $\mathcal{N}_{i,t}$.
**output** Models for deployment $\boldsymbol{w}_{i,T}$
    **for** $t = 0, 1, \ldots, T - 1$ **do**
        $\boldsymbol{w}_{i,t,0} = \boldsymbol{w}_{i,t}$
        **for** $k = 0, 1, \ldots, K - 1$ **do**
            $\boldsymbol{w}_{i,t,k+1} = \boldsymbol{w}_{i,t,k} - \eta \nabla f_i(\boldsymbol{w}_{i,t,k}; \xi)$
        **end for**
        Gossip $\boldsymbol{w}_{i,t,K}$ to her neighbours $\mathcal{N}_{i,t}$.
        $\boldsymbol{w}_{i,t+1} = \text{Avg}(\{\boldsymbol{w}_{i',t,K}\}_{i' \in \mathcal{N}_i} \cup \boldsymbol{w}_{i,t,K})$
    **end for**

---

**Network topology and aggregation.** Before aggregation, the clients would transmit their gradient update to their neighbors. The set of neighbors that a peer is defined by the *Network topology*. We in this paper simulate three main kinds of topology, which is visually displayed in Figure 8. Once receiving models from her neighbors, each client aggregates the models by directly averaging them, and subsequently produces the model used for the next round of local training.

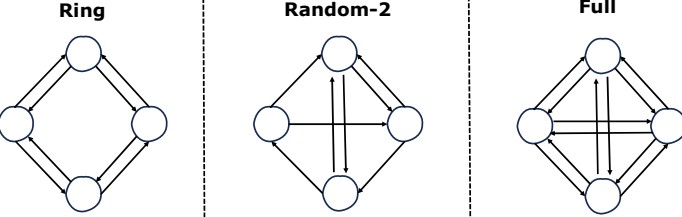

Figure 6: Common network topology in DFL. For Ring topology, each client would only send and receive models from her immediate neighbours. For Random topology, each client in each round will send and receive models from a specific number of random peers. For Full topology, each client will broadcast and receive her models from all the other clients in the system.

## A.2 Taxonomy of existing backdoor defense in FL

We now have a more comprehensive discussion on existing backdoor/data poisoning defense in FL.

**Certified robustness**. This category of study aims at developing a model (or an ensemble of models) that enjoys higher robustness to backdoor attacks. More specifically, they develop certified robustness bound such that they prove the deployed model is not sensitive to input perturbation – the classification of input would not change if only bounded perturbation of the input. To achieve this goal, a smooth classifier with random noise added in the input (Alfarra et al., 2022), the weights (Xie et al., 2021), or a random ensemble of classifier (Cao et al., 2022) is utilized. However, existing study on certified robustness typically introduces larger computation overhead in the deployment stage, an ensemble of models, or an "ensemble" of noisy input is needed to complete an effective query. This property is undesirable because federated learning models are mostly deployed in a resource-constraint scenario.

**Adversarial training**. This category of study (Zizzo et al., 2020; Shah et al., 2021; Zhang et al., 2023; 2022a) re-formulates the optimization problem in FL as a min-max problem. In the inner maximization problem, the defender tries to optimize the adversarial samples that lead to the malfunction of the models, while in the outer minimization problem, it improves the model's robustness against the generated adversarial samples. However, due to the introduction of inner maximization problem, the training overhead needs to be doubled. This is undesirable in a resource-constraint DFL scenario.

**Robust aggregation**. A large body of federated backdoor defenses fall into this category. Robust aggregation is concerned with how to mitigate or eliminate the impact of poisoned local models in the aggregation phase. The general idea is to detect the poisoned local models and preclude them from aggregation into the global model. To achieve this goal, different detection criteria are applied, e.g., the spatial property (Tolpegin et al., 2020), spatial+ temporal properties (Chow et al., 2023), proximity to the cluster centroid (Blanchard et al., 2017), proximity to a benign proxy model (Cao et al., 2020), coordinate-wise consistency (Ozdayi et al., 2021), classification performance validation (Andreina et al., 2021; Guo et al., 2021), differences between the structure and outputs of models (Rieger et al., 2022), layer wise anomaly score (Lin et al., 2023), spectral anomaly scores (Li et al., 2020b), loss difference of representatives between current and previous model(Elhattab et al., 2023), or the consistency of predicted model update (calculated from the estimated Hessian) and the received one (Zhang et al., 2022c). In addition to filtration of poisoned update, it is shown in (Nguyen et al., 2022) that robust aggregation can be combined with gradient clipping and noise adding to promote performance. However, this category of study generally exhibits poor performance in Non-IID settings (Li et al., 2023), because the heterogeneous local models can be easily viewed as outliers, which opens up an opportunity for the poisoned models to evade detection and still be able to aggregate into local models. Moreover, this category of study is vulnerable to adaptive attacks – attackers could readily adjust the model weights (probably by adding a regularizer in the training stage, like the Wasserstein Regularization in Doan et al. (2021)) such that the model does not exhibit the suspicious property, thereby evading detection.

There are a few defenses that cannot be categorized into the three broad defense categories, e.g., (Panda et al., 2022), which utilizes the existing gradient sparsification technique to craft a defense, and (Wu et al., 2020), which applied a pruning-finetuning technique (Liu et al., 2018) into federated learning context. There are also other techniques e.g., trigger-reversing (Wang et al., 2019; Liu et al., 2019; Wang et al., 2022; Liu et al., 2022), finetuning (Zhu et al., 2023; Huang et al., 2022), knowledge distillation (Pang et al., 2023; Li et al., 2021b; Wu et al., 2022), meta neural analysis (Xu et al., 2021; Kolouri et al., 2020; Cai et al., 2022) and data splitting (Gao et al., 2023) and augmentation (Qiu et al., 2021; Borgnia et al., 2021). All of them are backdoor defenses originally proposed in centralized setting, but might potentially be adapted to DFL setting. In this paper, we try to develop new category of defense among general defenses by merging the ideas from pruning-aware training.

## A.3 Discussion on client drift/communication reduction issue

In DFL setting, we take several local steps in the client optimization process, which significantly degrades the FL performance due to client drift issue (Zhao et al., 2018). To mitigate this issue, a plethora of work from an optimization standpoint has been proposed. For example, proximal-point based solution FedProx Li et al. (2018) corrects the local drift by adding a proximal term towards global model in the local training loss. Variance-reduction technique (Karimireddy et al., 2020) mitigate this issue by adding central control variates to correct the local drift. ADMM-based solution

(Zhang et al., 2020; Acar et al., 2021) strengthen the local consistent by introducing a auxiliary dual variable in the local training process. However, because Silencer also modifies the local procedure by doing pruning-aware training. Existing techniques for mitigating client-drift may not be directly integrate into Silencer, which is a fundamental research problem that is of interest. For example, when extending SCAFFOLD (Karimireddy et al., 2020), it is possible that the original design central variate cannot correct each client's client drift as all the clients are training on a subset of the parameters, and may result in very heterogeneous gradient. It is essential to design new control variate based on the consensus or the overlap of the gradient coordinate, which might alone be complicated enough. The same issue is faced when integrating dynamic regularization as proposed FedDyn (Acar et al., 2021). When the local procedure of every client is modified to train on a subset of parameters, directly applying dynamic regularization into local loss may not enable the server to reconstruct the global gradient, and therefore the server may not able to correct the client drift with the derived global statistic. We leave in the future work for further optimizing Silencer by integrating the existing solutions of client drift control.

Communication reduction (Konečnỳ et al., 2016) is also a fundamental research problem in FL. Existing solutions, e.g., 5GCS (Grudzień et al., 2023), ProxSkip-VR (Malinovsky et al., 2022), CompressedScaffnew (Condat et al., 2022) FedRR(Mishchenko et al., 2022), and Q-RR(Sadiev et al., 2022) aim to jointly optimize communication reduction/client drift issue. How to integrate these solutions for FL, or a modern error feedback technique (EF21 (Richtárik et al., 2021), EF21-P (Gruntkowska et al., 2023)) to further reduce communication of Silencer is also an important research agenda, which we would like to explore in the future.

## A.4 DISCUSSION ON PRIVACY ISSUE

It is shown in (Geiping et al., 2020; Wei et al., 2020b) that gradient-inversion attack can be applied to federated learning to reconstruct the training data. To further strengthen privacy protection, two main categories methods and their variants are typically adopted, i.e., secure aggregation (Bonawitz et al., 2017; Kadhe et al., 2020; So et al., 2022) and differential privacy (Wei et al., 2020a; Truex et al., 2020; Triastcyn & Faltings, 2019) For DP, the privacy-preserving method can be easily applied to Silence. We can just ask the clients to add DP noise to their model before gossiping gradient update. For secure aggregation, it is shown in (Jeon et al., 2021) that secure aggregation can also be adpated to DFL setting but needs some necessary modification. With Silener protocol, we may need further modification, which itself is an interesting research problem. We put this into our future agenda.

## A.5 DISCUSSION OF TYPE OF BACKDOOR ATTACK

Table 7: Performance comparison of dirty-label and clean-label attack in FL.

| Methods (Non-IID) | Benign Acc | ASR |
|---|---|---|
| DFedAvg (clean label) | 89.65 | 39.66 |
| DFedAvg (dirty label) | 88.70 | 92.56 |
| Silencer (clean label) | 86.71 | 3.75 |
| Silencer (dirty label) | 87.52 | 10.99 |

**Dirty-label backdoor vs. Clean-label backdoor.** Backdoor attack can be mainly classified into two types. The first type is dirty-label backdoor Gu et al. (2017), in which the attacker would poison data with arbitrary label in the training dataset by adding trigger and modifying label. The second type is clean-label backdoor, in which the attackers are not allowed to modifying the label but are only allowed to add trigger to the image of the target label. The motivation is that by modifying the label of the data, the poisoned data with inconsistent label can be easily filter out by a data filtration process. It is shown in (Turner et al., 2018) that clean label attack is less effective compared to dirty label attack in the centralized setting. We show in Table 7 that this is also true for federated learning. In order to compensate the performance in clean-label backdoor attack, (Turner et al., 2018) proposes to add adversarial noise or utilize a Generative adversarial network (GAN) to perturb the feature of the original samples. Subsequent study (Zeng et al., 2022) show that better attack performance can be gained by utilizing a synthesis trigger, which facilitate a better alignment with the target class. In federated learning setting, which is studied by this paper, we mainly simulate dirty-label backdoor attack Gu et al. (2017). This is because we assume the attackers have full control of the local device,

and therefore they can disable the data filtering process, which allows them to perform dirty-label backdoor without any constraints.

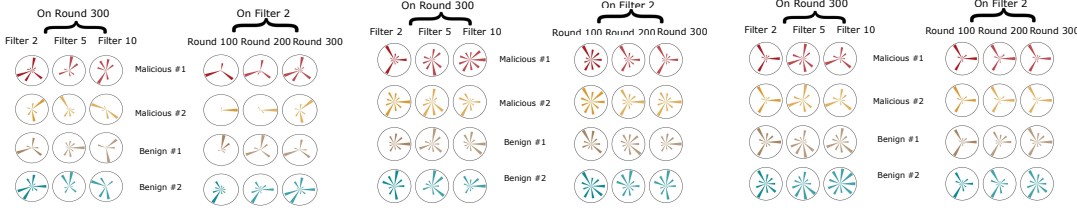

(a) Against clean-label backdoor. (b) On federated learning. (c) Against invisible backdoor.

Figure 7: Fisher information pattern for diversified setting (On FashionMNIST with a CNN). We show that: i) for two malicious clients, their last round FI patterns on the same filter are similar (client-level invariance) , and that ii) the FI pattern of malicious clients would not change much accross rounds(round-level invariance).

**Fisher information invariant pattern on clean-label backdoor.** Though clean-label backdoor attack is less serious in federated learning, it is interesting to study if the Fisher information invariant pattern also hold for clean label attack. To justify this, we show the pattern of a global model poisoned by clean-label attack in Figure 7a. Our results show that client-level and temporal-level invariance do not hold when evaluating clean-label backdoor. Our explanation for this is that the parameters for classifying backdoor trigger has not been developed well, which breaks the invariant pattern, and also leads to a reduced ASR.

**Fisher information invariant pattern on Federated learning.** Another concern is that can the Fisher information invariant pattern generalized to federated learning? For evaluation, we follow the same dirty-label attack setting in the main experiment, but utilize a server for centralized aggregation. In Figure 7b we show that the answer is affirmative.

**Fisher information invariant pattern on invisible trigger.** We in the main text show that the invariant pattern works for dirty-label backdoor attack with simple trigger, e.g., a "yellow plus sign". Advanced attacks, e.g., WaNet(Nguyen & Tran, 2021) , ISSBA (Li et al., 2021c) utilizes invisible trigger that is blended in the semantic of the victim data. We show in Figure 7c that WaNet is still not able to break the invariant pattern, though it is shown in their paper that WaNet can escape human inspection, reverse-engineer-based technique (Wang et al., 2019), neuron response screening (Liu et al., 2018), and perturbation-based screening (Gao et al., 2019). Though the invariant pattern seems to be attractive in judging if a backdoor model is poisoned or not, we need to acknowledge that this pattern is hard to be utilized in a centralized backdoor setting, because it relies on the unique poisoning structure (few poisoning datasets with high volumn of poisoned data) in federated learning.

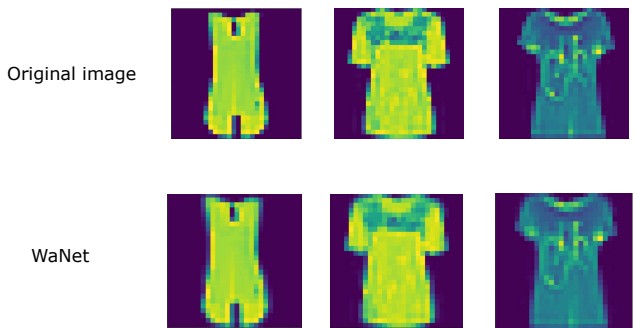

Figure 8: Visualization of invisible backdoor WaNet. As shown, the backdoor trigger planted by WaNet can not be distinguished from human eyes.

## B  DESCRIPTION OF SIMULATED BASELINES

We in this section provide a brief introduction of the simulated baseline method originally proposed in FL. Then we describe their implementation/adaption to the DFL setting.

### B.1 DEFENSE BASELINES

As the backdoor defense baselines in decentralized federated learning are not sufficient, we modified three representative defense baselines originally for conventional FL and applied them into our DFL setting. As follows, we briefly discuss their core idea and our adaptation into the DFL setting (please also consider checking our code for details).

**D-RLR.** RLR (Ozdayi et al., 2021) is the first and is also one of the most representative defenses specifically for backdoor attacks in FL. RLR is applied in the sever aggregation stage by inversing the sign of gradient update in those coordinates that have less sign consistency among clients. In our implementation tailored for DFL, we ask every client to take the sign inversion step when aggregating their neighbors' updates into their model. The hyper-parameter for the sign consistency threshold is set to 1/4 of the number of neighbors.

**D-Krum.** Krum (Blanchard et al., 2017) is first proposed to counter the Byzantine failures in FL, and later becomes a popular baseline for backdoor attack Zhang et al. (2022d); Xie et al. (2019). Krum is also applied in the aggregation phase, which aims to find the $i^*$-th client that minimizes $s(i) = \sum_{i \to j} \|V_i - V_j\|^2$ where $i \to j$ denotes the set of i-th client's $n - f - 2$ closest neighbors, and $V_i$ denotes the gradient update from client $i$. The gradient update $V_i^*$ is then applied in the global model, aiming to filter out the malicious update as far as possible. For D-Krum, we simply modify the Krum to each client's aggregation phase.

**D-FLTrust.** FLTrust (Cao et al., 2020) is another Byzantine-robust aggregation strategy applied in the FL's server aggregation phase. The core idea is to assign every gradient update a trust score based on its similarity with that of a benign surrogate dataset. The gradient to be applied is then the weighted average of gradient updates based on the trust score. To apply in DFL, we ask each client to use their local dataset as the surrogate dataset, compute the trust scores of each received gradient update, and then weighted average of the gradient updates.

**D-Bulyan**. Bulyan (Guerraoui et al., 2018) improves Krum by introducing trimmed mean after the multi-Krum operation. Similar to D-Krum, we adapt the Bulyan aggregation into each clients' aggregation phase.

### B.2 ATTACK BASELINES

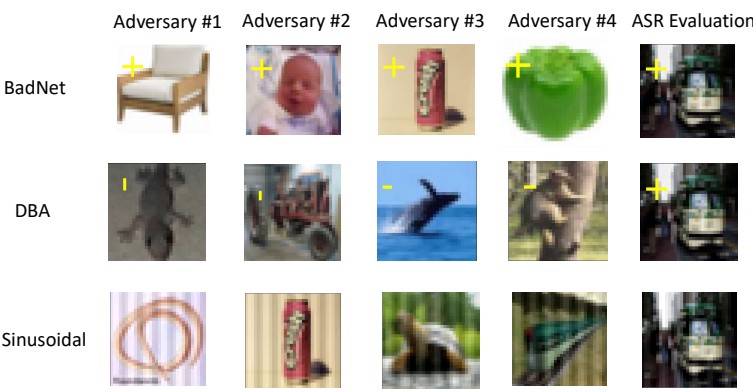

Figure 9: Examples of BadNet, DBA, and Sinusoidal attack. Labels of backdoor samples are modified to an arbitrary target label.

**BadNet**. BadNet Gu et al. (2017) is the first backdoor attack method proposed in the context of centralized machine learning. Later it is empirically proved in Bagdasaryan et al. (2020); Sun et al. (2019) that BadNet is also applicable in a federated learning context as long as a minority of clients' datasets are compromised.

**DBA**. DBA Xie et al. (2019) is a backdoor attack method that specifically targets federated learning. By decomposing a global trigger pattern into separate local patterns and embedding them into the training dataset of different adversaries within the system, the attack success rate of backdoor attack can be substantially higher and also can be more stealthy to evade the detection of several robust

aggregation schemes. In our implementation, we split the triggers into four different components, and each adversary would inject one component of them into their local dataset.

**Sinusoidal**. Instead of using a patch-like trigger, Sinusoidal (Barni et al., 2019) interposes a horizontal sinusoidal signal defined by $v(i, j) = \Delta \sin(2\pi j f/m)$, $1 \le j \le m, 1 \le i \le l$, for a certain frequency $f$, where $i$ and $j$ correspond to one pixel in the image. It is claimed that such a trigger pattern is more stealthy against human inspection but still can be recognized by a neural network.

**Neurotoxin**. Under the case that the adversaries can only participate in the early rounds of training, the model poisoned by the vanilla BadNet method would be gradually purified by the later stage of training over non-poisoned data. To increase the durability of the attack, (Zhang et al., 2022d) proposes Neurotoxin, whose basic idea is to ask the adversaries to project their poisoned gradient update to weight coordinates that have less chance to be updated by the benign clients (i.e., those coordinates that have smaller absolute serer update value). In our implementation in the DFL setting, we ask each adversary to collect the gradient updates from their neighbors and ask them to project their poisoned update to the coordinates that have the smallest absolute neighbor's update.

## C  DESIGN OF SILENCER AND ITS ADAPTIVE ATTACKS

We in this section subsequently describe the high-level idea of Silencer and the detailed implementation of some of its main components. Then we describe two adaptive attacks that assume the knowledge of Silencer defense and try to break it. Based on the insights from the adaptive attack design, we incorporate two variants of Silencer defense to strengthen protection or reduce unnecessary communication. Finally, we provide a security analysis to pinpoint the blind spot of our defense.

### C.1  HIGH-LEVEL IDEA OF SILENCER

We now summarize the high-level idea of Silencer for better understanding.

- **Function of mask searching.** In joint mask searching and pruning-aware training process, Silencer asks clients to find the coordinates important to their local data and include them in their masks.

- **Function of pruning-aware/sparse training.** Then it asks each client to only train a sparse core model as identified by their masks (i.e., only positions with values that are 1 in the corresponding mask are trained).

- **Consensus filtering**. Because the poisoned parameters are not important to the benign clients, which account for the majority. We can filter out those poisoned parameters by looking at the degree of overlap of the masks. Those more "popular" coordinates are less likely to be the poisoned ones, while those coordinates that only appear in a few client's active parameter sets are most likely to be the poisoned ones. Motivated by this thought, we propose consensus filtering to filter out those suspected poisoned parameters and recover the model. Note that we do not do filtering in each federated round, but do it after training has finished.

Another clarification we want to make is:

> *Why pruning-aware training induces less benign accuracy loss, compared to the pure pruning baselines (e.g., FIP-C)?*

This is because the coordinates reserved after consensus filtering will have a large overlap with the sparse models being trained by the benign clients in the training phase, and therefore the activation of the model would not experience significant change after consensus filtering. In other words, the benign clients are aware of pruning afterward, and try to train the sparse core model instead of the full model, which mitigates the performance loss due to pruning of dummy/malicious parameters.

### C.2  DETAILED IMPLEMENTATION OF SILENCER

**Mask initialization.** For mask initialization, we enforce different sparisty to different layers of the model in order to boost the sparse model's practical performance. We use Erdős–Rényi Kernel (ERK) (Evci et al., 2020) to distribute sparsity to different layers of a model. Specifically, the active

parameters of the convolutional layer initialized by ERK are proportional to $1\frac{n^{1-1}+n^l+w^l+h^l}{n^{l-1}*n^l*w^l*h^l}$, where $n^{l-1}$, $n_l$ $w^l$ and $h^l$ respectively specify the number of input channels, output channels and kernel's width and height in the $l$-th layer. For the linear layer, the number of active parameters scales with $1\frac{n^{l-1}+n^l}{n^{l-1}*n^l}$ where $n^{l-1}$ and $n^l$ are the number of neurons in the layer $(l-1)$ -th and $l$ -th. In essence, this mask initialization prunes more aggressively on those layers that have more parameters while being more lenient to those layers that intrinsically have fewer parameters.

**Decay pruning rate with cosine annealing** In our mask pruning/reclamation process, we let clients prune out $\alpha_t$ percentage of parameters and recover the same amount of parameters. In order to control the extent of mask evolution, and to make the mask of each client stable after a sufficient amount of training rounds, the pruning parameter $\alpha_t$ is set to be decayed with the initial rate $\alpha_0$ with cosine annealing, which can be formalized as follows:

$$\alpha_t = 0.5 \times \alpha_0 \times \left(1 + \cos\left(\frac{t}{T_{end}}\pi\right)\right) \tag{10}$$

where $t$ is the number of communication round, and $T$ is the number of rounds we run the algorithm.

**Pruning/Reclamation.** We enforce an alternating pruning/reclamation process in order to maintain the same sparsity of the model throughout training. This design is in sharp contrast to literature in neural network pruning e.g., (Li et al., 2020a; Frankle & Carbin, 2018). Our design motivation is based on the observation that the pure pruning technique is very sensitive to the chosen pruning rate – the model would easily collapse if the pruning ratio is not set properly, which results in unstable training. To achieve constant sparsity throughout training, in the pruning process, we should not prune the parameters that are already marked as prune, and the same for the reclamation process, i.e., we should not recover the parameters that are already active. For the sake of clearness, we present the PyTorch style code for our pruning/reclamation process in Algorithm 3. As shown, we obtain the TopK/BottomK diagonal of empirical FI only among the candidate set, but the sort values of those not qualified ones (actives ones in the reclamation process, and inactive one in the pruning process) are set to a sufficiently large/small value.

---

**Algorithm 3** Pruning/reclamation sub-procedure

---

**input** $F_i(\boldsymbol{w}_{i,t,0}), \boldsymbol{m}_{i,t}$
**output** Coordinates to be pruned: $\text{ArgBtmK}_{\alpha_t}(F_i(\boldsymbol{w}_{i,t,0}))$
    **for** $l = 0, 1, \ldots, L-1$ **do**
        $num_{prune} = \alpha_t \times$ # of params in the l-th layer
        $sort = \text{torch.where}(\boldsymbol{m}_{i,t}^{(l)} = 1, F_i(\boldsymbol{w}_{i,t,0}), 1000\times\text{torch.ones\_like}(F_i(\boldsymbol{w}_{i,t,0})))$
        $\_, idx = \text{torch.sort}(sort.\text{view}(-1))$
        $\boldsymbol{m}_{i,t+1}^{(l)}.\text{view}(-1)[idx[: num_{prune}]] = 0$
    **end for**

**input** $F_i(\boldsymbol{w}_{i,t,0}), \boldsymbol{m}_{i,t}$
**output** Coordinates to be recovered: $\text{ArgTopK}_{\alpha_t}(F_i(\boldsymbol{w}_{i,t,0}))$
    **for** $l = 0, 1, \ldots, L-1$ **do**
        $num_{prune} = \alpha_t \times$ # of params in the l-th layer
        $sort = \text{torch.where}(\boldsymbol{m}_{i,t+\frac{1}{2}}^{(l)} = 0, F_i(\boldsymbol{w}_{i,t,0}), -1000\times\text{torch.ones\_like}(F_i(\boldsymbol{w}_{i,t,0})))$
        $\_, idx = \text{torch.sort}(sort.\text{view}(-1), \text{descending=True})$
        $\boldsymbol{m}_{i,t+\frac{1}{2}}^{(l)}.\text{view}(-1)[idx[: num_{prune}]] = 1$
    **end for**

---

**Consensus filtering**. After training has finished, each client would do consensus filtering in order to remove the poisoned parameters that nested in their model. In order to perform such an operation, clients would need the mask information from all the other peers in the system. Therefore, in our implementation, once the training is finished, each client would start the stage-2 optimization by broadcasting their masks to all the peers in the system, regardless of the network topology. We argue that such an operation is communication-efficient because the masks are typically a 0-1 vector that does not require much communication overhead. In the case that the client can by no means reach all the other clients, extra design needs to be considered (e.g., requiring a truthful central coordinator, or a carefully designed peer-to-peer broadcast scheme). We leave discussion of this case to future work.

## C.3 ADAPTIVE ATTACKS TO SILENCER

We simulate three adaptive attacks that assume the knowledge of our defense and try to break it, which will be specified as follows.

---

**Algorithm 4** Adaptive attacker: FixMask

**input** Iteration $T$; Local steps $K$; Learning rate $\eta$; Pruning rate $\alpha_t$; Initial model $\boldsymbol{w}_{i,0}$; Neighbours, $\mathcal{N}_{i,t}$.
**output** Models for deployment $\tilde{\boldsymbol{w}}_{i,T}$
  Initialize mask $\boldsymbol{m}_{i,0}$ with overall sparsity $s$.
  **for** $t = 0, 1, \ldots, T-1$ **do**
    $\boldsymbol{w}_{i,t,0} = \boldsymbol{w}_{i,t} \odot \boldsymbol{m}_{i,0}$
    **for** $k = 0, 1, \ldots, K-1$ **do**
      $\boldsymbol{w}_{i,t,k+1} = \boldsymbol{w}_{i,t,k} - \eta \boldsymbol{m}_{i,0} \odot \nabla f_i(\boldsymbol{w}_{i,t,k}; \xi)$
    **end for**
    $\boldsymbol{w}_{i,t+\frac{1}{2}} = \boldsymbol{w}_{i,t} - \boldsymbol{m}_{i,0} \odot (\boldsymbol{w}_{i,t,0} - \boldsymbol{w}_{i,t,K})$
    Gossip $\tilde{\boldsymbol{w}}_{i,t+\frac{1}{2}}$ to her neighbours $\mathcal{N}_{i,t}$.
    $\boldsymbol{w}_{i,t+1} = \mathrm{Avg}(\{\tilde{\boldsymbol{w}}_{i',t+\frac{1}{2}}\}_{i' \in \mathcal{N}_i} \cup \boldsymbol{w}_{i,t+\frac{1}{2}})$
  **end for**
  Gossip masks and obtain $\{\boldsymbol{m}_{1,T}, \ldots, \boldsymbol{m}_{M,T}\}$
  Deploy $\tilde{\boldsymbol{w}}_{i,T} = \mathcal{Q}_\theta(\boldsymbol{m}_{1,T}, \ldots, \boldsymbol{m}_{M,T}) \odot w_{i,T}$

---

**Algorithm 5** Adaptive attacker: Curiosity

**input** Iteration $T$; Local steps $K$; Learning rate $\eta$; Pruning rate $\alpha_t$; Initial model $\boldsymbol{w}_{i,0}$; Neighbours, $\mathcal{N}_{i,t}$.
**output** Models for deployment $\tilde{\boldsymbol{w}}_{i,T}$
  Initialize mask $\boldsymbol{m}_{i,0}$ with overall sparsity $s$.
  **for** $t = 0, 1, \ldots, T-1$ **do**
    Get $\boldsymbol{m}^* = \mathcal{Q}_\theta(\boldsymbol{m}_{1,t}, \ldots, \boldsymbol{m}_{M,t})$ by some ways.
    $\boldsymbol{w}_{i,t,0} = \boldsymbol{w}_{i,t} \odot \boldsymbol{m}^*$
    **for** $k = 0, 1, \ldots, K-1$ **do**
      $\boldsymbol{w}_{i,t,k+1} = \boldsymbol{w}_{i,t,k} - \eta \boldsymbol{m}^* \odot \nabla f_i(\boldsymbol{w}_{i,t,k}; \xi)$
    **end for**
    $\boldsymbol{w}_{i,t+\frac{1}{2}} = \boldsymbol{w}_{i,t} - \boldsymbol{m}^* \odot (\boldsymbol{w}_{i,t,0} - \boldsymbol{w}_{i,t,K})$
    Gossip $\tilde{\boldsymbol{w}}_{i,t+\frac{1}{2}}$ to her neighbours $\mathcal{N}_{i,t}$.
    $\boldsymbol{w}_{i,t+1} = \mathrm{Avg}(\{\tilde{\boldsymbol{w}}_{i',t+\frac{1}{2}}\}_{i' \in \mathcal{N}_i} \cup \boldsymbol{w}_{i,t+\frac{1}{2}})$
  **end for**
  Gossip masks and obtain $\{\boldsymbol{m}_{1,T}, \ldots, \boldsymbol{m}_{M,T}\}$
  Deploy $\tilde{\boldsymbol{w}}_{i,T} = \mathcal{Q}_\theta(\boldsymbol{m}_{1,T}, \ldots, \boldsymbol{m}_{M,T}) \odot w_{i,T}$

---

**FixMask.** For FixMask attack, we assume the adversaries do not change the mask in the mask searching process, but keeps it unchanged as the initialized one. For other benign clients, they still strictly enforce the mask-searching process. Our results in Table 3 show that Silencer is robust to FixMask attack simply by properly enlarging the pruning/reclamation ratio. By doing so, the benign clients may reach a consensus mask that is substantially different from the initial one, thereby eliminating the risk of FixMask attack.

**Curiosity.** For Curiosity attack, we assume the attackers can figure out the masks of its neighbors by looking at the weights sent by them. This can be achieved if the adversary compares two models sent from one client over two sequential iterations – those coordinates that have less absolute update value should correspond to those that are masked out to 0 in the local training phase. After figuring out the masks of each client in the system, the adversaries produce the consensus mask and continuously project their poisoned update within it. Our results in Table 3 show that Silencer is not robust to Curiosity attack. This result inspires us to develop a security enhancement version of Silencer, which will be specified in the later section.

**BenignSearch.** For BenignSearch, we assume the attackers use their own non-poisoned dataset for mask searching. Then it uses its poisoned dataset (built upon the original benign dataset) to update the parameters within the active parameters (as specified by the mask). Our results in Table 3 show that Silencer is robust to FixMask attack also by properly enlarging the pruning/reclamation ratio.

## C.4 TWO VARIANTS OF SILENCER

We present two variants of Silencer respectively for security enhancement and communication reduction in this section. The two variants are to be applied in different security assumptions over adaptive adversaries within the system. The key properties of them are summarized in Table 8.

Table 8: Properties of Silencer and its variants.

| Variants | against dataset poison | against adaptive attack | comm reduction | inference reduction |
|---|---|---|---|---|
| Silencer | strong | medium | ✗ | ✓ |
| Silencer-SE | strong | strong | ✗ | ✓ |
| Silencer-CR | strong | weak | ✓ | ✓ |

**Silencer with Security Enhancement (Silencer-SE).** From Table 3, it can be found that Silencer is not robust to one of the adaptive attacks named curiosity. Curiosity is conducted by adaptive adversaries who try to infer the consensus mask from the model transmitted from their benign

neighbors. An intuitive idea to mitigate the mask information leakage is to ask all the benign clients to add noise to the model weights in the federated training stage. By doing this, the adversary may not deduce useful mask information from the noisy version of weights – it becomes difficult for them to know whether the change of her neighbors' weights results from the sparse local gradient update or the deliberately added noise. Given the intuition, we present the security enhancement variant of Silencer in Algorithm 6, where we ask each client (potentially only the benign ones) to sample noise from a zero-mean Gaussian distribution with noise intensity $\sigma^2$. Silencer-SE is suitable to be applied under the security assumption that the clients can actively infer the consensus, and can also modify its own local training protocol.

$$\boldsymbol{w}_{t+1} = \mathrm{Avg}(\{(\boldsymbol{w}_{i,t} - \boldsymbol{m}_{i,t+1} \odot (\boldsymbol{w}_{i,t,0} - \boldsymbol{w}_{i,t,K})) + \varepsilon_{i,t}\}_{i \in \mathcal{N}_i}) \qquad (11)$$

**Silencer with Communication Reduction (Silencer-CR).** Under the security assumption that the adversaries cannot modify the local training protocol, and their data are either intentionally modified by themselves or are unintentionally being polluted in the data collection phase, we propose Silencer-CR to further reduce communication between clients. As shown in Algorithm 7, Silencer-CR asks each client to only transmit the sparse gradient update to each other. After receiving the sparse gradient updates from neighbors, the updates are applied to the dense model individually owned by each client. Inevitably, this operation would ask each client to disclose its individual mask to all its neighbors, which pose a serious threat if one or a few of them are adaptive adversaries. Therefore, Silencer-CR *can only be applied when clients within the system are "partially trustful" in that they will not modify their local training protocol after knowing others' masks.*

---

**Algorithm 6** Silencer with security enhancement

**input** Iteration $T$; Local steps $K$; Learning rate $\eta$; Pruning rate $\alpha_t$; Initial model $\boldsymbol{w}_{i,0}$; Neighbours, $\mathcal{N}_{i,t}$.
**output** Models for deployment $\tilde{\boldsymbol{w}}_{i,T}$
  Initialize mask $\boldsymbol{m}_{i,0}$ with overall sparsity $s$.
  **for** $t = 0, 1, \ldots, T-1$ **do**
    $\boldsymbol{w}_{i,t,0} = \boldsymbol{w}_{i,t} \odot \boldsymbol{m}_{i,t}$
    Compute $F_i(\boldsymbol{w}_{i,t,0})$ per Eq. (4)
    $\boldsymbol{m}_{i,t+\frac{1}{2}} = \boldsymbol{m}_{i,t} - \mathrm{ArgBtmK}_{\alpha_t}(F_i(\boldsymbol{w}_{i,t,0}))$
    $\boldsymbol{m}_{i,t+1} = \boldsymbol{m}_{i,t+\frac{1}{2}} + \mathrm{ArgTopK}_{\alpha_t}(F_i(\boldsymbol{w}_{i,t,0}))$
    **for** $k = 0, 1, \ldots, K-1$ **do**
      $\boldsymbol{w}_{i,t,k+1} = \boldsymbol{w}_{i,t,k} - \eta \boldsymbol{m}_{i,t+1} \odot \nabla f_i(\boldsymbol{w}_{i,t,k}; \xi)$
    **end for**
    $\boldsymbol{w}_{i,t+\frac{1}{2}} = \boldsymbol{w}_{i,t} - \boldsymbol{m}_{i,t+1} \odot (\boldsymbol{w}_{i,t,0} - \boldsymbol{w}_{i,t,K})$
    $\tilde{\boldsymbol{w}}_{i,t+\frac{1}{2}} = \boldsymbol{w}_{i,t+\frac{1}{2}} + \varepsilon$ where $\varepsilon$ is a sample from a zero-mean Gaussian with variance $\sigma^2$.
    Gossip $\tilde{\boldsymbol{w}}_{i,t+\frac{1}{2}}$ to her neighbours $\mathcal{N}_{i,t}$.
    $\boldsymbol{w}_{i,t+1} = \mathrm{Avg}(\{\tilde{\boldsymbol{w}}_{i',t+\frac{1}{2}}\}_{i' \in \mathcal{N}_i} \cup \boldsymbol{w}_{i,t+\frac{1}{2}})$
  **end for**
  Gossip masks and obtain $\{\boldsymbol{m}_{1,T}, \ldots, \boldsymbol{m}_{M,T}\}$
  Deploy $\tilde{\boldsymbol{w}}_{i,T} = \mathcal{Q}_\theta(\boldsymbol{m}_{1,T}, \ldots, \boldsymbol{m}_{M,T}) \odot w_{i,T}$

---

**Algorithm 7** Silencer with comm reduction

**input** Iteration $T$; Local steps $K$; Learning rate $\eta$; Pruning rate $\alpha_t$; Initial model $\boldsymbol{w}_{i,0}$; Neighbours, $\mathcal{N}_{i,t}$.
**output** Models for deployment $\tilde{\boldsymbol{w}}_{i,T}$
  Initialize mask $\boldsymbol{m}_{i,0}$ with overall sparsity $s$.
  **for** $t = 0, 1, \ldots, T-1$ **do**
    $\boldsymbol{w}_{i,t,0} = \boldsymbol{w}_{i,t} \odot \boldsymbol{m}_{i,t}$
    Compute $F_i(\boldsymbol{w}_{i,t,0})$ per Eq. (4)
    $\boldsymbol{m}_{i,t+\frac{1}{2}} = \boldsymbol{m}_{i,t} - \mathrm{ArgBtmK}_{\alpha_t}(F_i(\boldsymbol{w}_{i,t,0}))$
    $\boldsymbol{m}_{i,t+1} = \boldsymbol{m}_{i,t+\frac{1}{2}} + \mathrm{ArgTopK}_{\alpha_t}(F_i(\boldsymbol{w}_{i,t,0}))$
    **for** $k = 0, 1, \ldots, K-1$ **do**
      $\boldsymbol{w}_{i,t,k+1} = \boldsymbol{w}_{i,t,k} - \eta \boldsymbol{m}_{i,t+1} \odot \nabla f_i(\boldsymbol{w}_{i,t,k}; \xi)$
    **end for**
    $\boldsymbol{g}_{i,t} = (\boldsymbol{w}_{i,t,0} - \boldsymbol{w}_{i,t,K})$
    Gossip $\boldsymbol{g}_{i,t}$ to her neighbours $\mathcal{N}_{i,t}$.
    $\boldsymbol{w}_{i,t+1} = \boldsymbol{w}_{i,t} + \frac{1}{M} \sum_{i=1}^{M} \boldsymbol{g}_{i,t}$
  **end for**
  Gossip masks and obtain $\{\boldsymbol{m}_{1,T}, \ldots, \boldsymbol{m}_{M,T}\}$
  Deploy $\tilde{\boldsymbol{w}}_{i,T} = \mathcal{Q}_\theta(\boldsymbol{m}_{1,T}, \ldots, \boldsymbol{m}_{M,T}) \odot w_{i,T}$

---

## C.5 SECURITY ANALYSIS

We identify two potential attack surfaces of Silencer, and separately discuss as follows.

**Can Silencer be secure against adaptive attacks in the training stage?** As shown in the previous section, vanilla Silencer at least cannot successfully defend against one of the adaptive attacks, which we name Curiosity. However, to perform Curiosity attack, additional information on the masks of its neighbours is required to obtain in prior. This information is typically hard to infer, and it could be even harder if the mask information is encrypted by enforcing a security-enhanced version of Silencer (Silencer-SE). Moreover, there are other potential hardware-based security-enhanced techniques, e.g., TEE (Mo et al., 2021) that can enforce the same training procedure for all the clients, which precludes the possibility of conducting adaptive attacks by some adversaries.

**Can Silencer be secure against adaptive attacks in consensus filtering stage?** In our evaluation, we design two adaptive attacks to demystify the vulnerability of Silencer in the local training phase. However, in the consensus filtering stage, we require each client to faithfully broadcast their masks to all the other clients, which may render a potential attack surface for the adversaries. We insist that it is extremely hard, if not impossible, for adversaries to introduce a backdoor by solely faking their mask at this stage. Our claim is justified by two reasons: i) as the adversaries only constitute a small portion of clients, and each of them only holds one vote in the consensus filtering, they actually cannot significantly modify the consensus mask. ii) the adversaries have no knowledge of other client's masks within the mask broadcast phase before it is finished, which poses great obstacles for them to make a good attack decision. Currently, we cannot develop an effective adaptive attack solely in the consensus filtering stage, and we leave the study of the adaptive attacks as future works.

# D  EXPERIMENT SETUP AND MISSING RESULTS

## D.1  DETAILED SETUP

**Datasets and augmentation.** We use FashionMnist, GTSRB CIFAR10/CIFAR100, TinyImagenet to benchmark the algorithms. For FashionMnist, the images have a resolution of $28 \times 28$ with 10 labels. For GTSRB, the images' resolution is $30 \times 30$, but we reshape it to $32 \times 32$ in order to feed into a ResNet-9 model. In CIFAR10 dataset, there are 50,000 training images, and 10,000 testing images, which contain 10 labels. Each sample has the resolution of $32 \times 32$. In CIFAR100, we have the same amount of training and testing, but it has a total number of 100 labels. For TinyImagenet, the original resolution is $64 \times 64$, we don't reshape it to $32 \times 32$ because reshaping may lose accuracy. Instead, we modify the ResNet-9 model by adding a pooling layer after its first convolutional layer to keep the same hidden size. For augmentation, we use RandomCrop+ Random Horizontal Flip for CIFAR10/100 to mitigate overfitting. For GTSRB and TinyImagenet, we use only RandomCrop and for FashionMnist we don't use augmentaion. For GTSRB, CIFAR10/100, and TinyimageNet, we use a ResNet9 model, but for FashionMnist we use a simple 5-layer CNN because this task is easier. For the simulation of Non-IID, we use Dirichlet distribution (Hsu et al., 2019) for simulation, with default parameter 0.5. We summarize the dataset setup in Table 9.

Table 9: Dataset setup.

| Datasets | Data size(train/test) | Resolution | Reshape Resolution | Augmentation | Model |
|---|---|---|---|---|---|
| FashionMnist | 60,000/ 10,000 | $28 \times 28$ | $32 \times 32$ | Normalized | CNN |
| GTSRB | 34,799/ 12,630 | $30 \times 30$ | $32 \times 32$ | RandomCrop+Normalized | ResNet9 |
| CIFAR10 | 50,000/10,000 | $32 \times 32$ | - | RandomCrop+Horizon Flip+Normalized | ResNet9 |
| CIFAR100 | 50,000/10,000 | $32 \times 32$ | - | RandomCrop+Horizon Flip+Normalized | ResNet9 |
| TinyImagenet | 100,000/10,000 | $64 \times 64$ | - | RandomCrop+Normalized | modified ResNet9 |

**Hyperparameter Selection**. For general training hyper-parameters, we pick the best local learning rate from $[0.1, 0.01, 0.001]$, weight decay parameter from $[1e - 3, 1e - 4]$, local epochs from $[1, 2, 3]$, and batch size from $[64, 128]$ by running FedAvg algorithm on CIFAR10 for 300 rounds. The best grid-searching result is $(0.1, 1e - 4, 2, 64)$, which we adopt in evaluation. We keep this default setup for all five datasets because of the limitation of resources. We show the general setting in Table 10.

Table 10: General hyper-parameter selection.

| Hyper-parameter | Selection | Chosen value |
|---|---|---|
| learning rate | [0.1,0.01, 0.001] | 0.1 |
| weight decay | [1e-3, 1e-4] | 1e-4 |
| local epochs | [1,2,3] | 2 |
| batch size | [64,128] | 64 |

For defense-specific hyper-parameters, we select based on the following criterion. The threshold for D-RLR is chosen from [10%, 25%, 50%, 75% ] of neighbor size, and we choose the optimal one, i.e., 25%. The reason we associate it with neighbor size is that the number of models participates in aggregations vary for topology with different neighbor size. For D-Krum, we use the original Krum implementation but not mulit-krum, i.e., in each step we only select one gradient from the aggregation list. The outlier number $m$ is set to be $min(\text{\# of attackers} , \text{neighbours size})$ based on

the same reason we show for D-RLR. For FLTrust, there is not an algorithm-specific hyper-parameter. Each client uses its local dataset as the root dataset. For D-Bulyan, we choose the multi-Krum parameter as 3, and the trimmed mean parameter as 1, respectively chosen from [3,4,5] and [1,2]. For Silener, we have three hyper-parameters. For sparsity $s$, we choose from [0.2,0.5,0.75,0.9]. For initial pruning/reclamation rate $\alpha_0$, we choose from [1e-6, 1e-4,1e-3, 1e-2]. For consensus threshold $\theta$, we choose from [5,10,15,20,25,40]. The default setting is the best one on CIFAR10, which is (0.75, 1e-4,20). We summarize the selection in Table 11.

Table 11: Defense specific hyper-parameter selection.

| Hyper-parameter | Selection | Chosen value |
|---|---|---|
| threshold for D-RLR | [10%,25%, 50%, 75%] of neighbour size | 25% |
| outlier number for Krum | - | $min$(# of attackers , neighbours size) |
| multi-Krum parameter for Bulyan | [3,4,5] | 3 |
| trimmed mean parameter for Bulyan | [1,2] | 1 |
| sparsity $s$ for Silencer | [0.2,0.5,0.75,0.9] | 0.75 |
| initial pruning/reclamation rate for Silencer | [1e-6, 1e-4,1e-3, 1e-2] | 1e-4 |
| Consensus threshold for Silencer | [5,10,15,20,25,40] | 20 |

For the attack setting, we use BadNet as the default attack method. We use a yellow plus sign as a backdoor trigger (see Figure 9 for example). For sinusoidal, we choose the intensity and frequency to 20 and 6 respectively, which is the default hyper-parameter in their paper (Barni et al., 2019). For Neurotoxin, we choose its density from [0.25,0.5,0.75] on CIFAR10 and choose the best one, which is 0.25. For all the attacks, we choose the 7th label as the target label, and *there is not source label*.

## D.2 MISSING EXPERIMENTS RESULTS

We present our experimental results that cannot fit into the main text here.

**ASR across clients.** Figure 10 shows the ASR of the 40 clients under D-FedAvg without defense. For Ring topology, the models of those clients who are neighbors of adversaries are severely poisoned, while those far away from the compromised clients remain benign and not infected. However, for random topology – the model of every client in the system is poisoned due to the dynamics of the P2P topology (i.e., frequently changing neighbors).

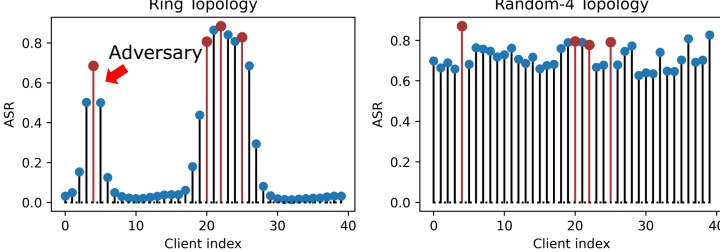

Figure 10: ASR across clients with no defense.

**Defense against advanced attack methods.** We simulate more advanced attacks DBA, sinusoidal, and Neurotoxin in Table 12. Overall, Silencer maintains superior defense ($< 10\%$ ASR) against all the advanced attacks while remaining comparable benign accuracy.

**Noisy Silencer with different noise intensity.** We show that Silencer is vulnerable to the case when the adversary can fully control the local training protocol and knows the consensus mask. To mitigate the risk of exposing the consensus mask, we ask every client to add random noise to the communicated model (See Silencer-SE in Appendix C.4). We show in Table 13 how noise intensity impacts the performance. Our conclusion is that a larger noise will degrade the benign accuracy, but can increase the robustness of the adaptive attack by making it harder to infer the consensus mask.

## D.3 ABLATION STUDY

We now provide the missing results of our ablation study to study the necessity of each essential component of Silencer. The component adopted by Silencer is marked in a gray background.

Table 12: Silencer on advanced attack. Blue number is the baseline FedAvg without defense.

| Methods (IID) | Benign Acc(%) ↑ | ASR(%) ↓ | Backdoor Acc(%) ↑ |
|---|---|---|---|
| BadNet | 91.04 (91.42) | 2.67 (75.09) | 87.22 (23.66) |
| DBA | 90.30 (91.79) | 3.02 (53.51) | 86.64 (50.18) |
| Sinusoidal | 90.30 (91.18) | 4.16 (14.72) | 85.23 (80.61) |
| Neurotoxin | 90.14 (91.57) | 5.12 (73.09) | 84.29 (33.27) |
| Methods (Non-IID) | Benign Acc(%) ↑ | ASR (%) ↓ | Backdoor Acc(%) ↑ |
| BadNet | 87.48 (89.08) | 3.63 (86.58) | 82.01 (12.52) |
| DBA | 87.18 (89.61) | 2.69 (80.10) | 81.71 (22.20) |
| Sinusoidal | 87.73 (89.06) | 3.23 (21.34) | 81.78 (70.79) |
| Neurotoxin | 86.46 (89.42) | 2.45 (77.30) | 81.81 (23.18) |

Table 13: Silencer-SE on noise intensity.

| Noise intensity (IID) | Benign Acc(%) ↑ | ASR(%) ↓ | Backdoor Acc(%) ↑ |
|---|---|---|---|
| 0.1 | 85.85 | 2.24 | 83.82 |
| 0.01 | 89.90 | 3.25 | 85.80 |
| 0.001 | 90.32 | 3.23 | 85.81 |
| 0.0001 | 90.08 | 3.46 | 85.97 |
| 0 | 91.04 | 2.67 | 87.22 |
| Noise intensity (Non-IID) | Benign Acc(%) ↑ | ASR (%) ↓ | Backdoor Acc(%) ↑ |
| 0.1 | 78.90 | 2.18 | 74.52 |
| 0.01 | 85.50 | 2.24 | 79.01 |
| 0.001 | 87.07 | 2.20 | 80.70 |
| 0.0001 | 86.52 | 2.93 | 80.74 |
| 0 | 87.48 | 3.63 | 82.01 |

**ERK initialization.** In Silencer, we utilize layer-wise sparsity to ensure practical performance. Here we replace the ERK initialization to Uniform initialization, which uniformly assigns a fixed sparsity to each layer in the model. Table 14 demonstrates the necessity of ERK initialization, which significantly boosts benign accuracy (over 5% of accuracy enhancement) and lowers ASR (over 4% of reduction), compared to the Uniform initialization.

Table 14: Ablation study for sparsity initialization.

| Methods (IID) | Benign Acc ↑ | ASR ↓ | Backdoor Acc ↑ |
|---|---|---|---|
| Uniform | 83.17 | 6.65 | 76.21 |
| ERK (ours) | **91.04** | **2.67** | **87.22** |
| Methods (Non-IID) | Benign Acc ↑ | ASR ↓ | Backdoor Acc ↑ |
| Uniform | 82.03 | 4.08 | 76.70 |
| ERK (ours) | **87.48** | **3.63** | **82.01** |

**Fisher guidance in pruning/reclamation.** In our pruning/reclamation procedure, we use the diagonal of empirical FI as an indicator of coordinates to be pruned and to be recovered. We show in Table 15 our ablation study on the way to do pruning/reclamation. We compare our original design with random pruning/reclamation, which randomly selects coordinates to be pruning/reclamation. Our results show that the guidance of FI is important, especially for the reclamation process, as replacing it with random selection would lead to catastrophic failure of the Silencer.

Table 15: Ablation study for parameters reclamation implementation.

| Methods (IID) | Benign Acc(%) ↑ | ASR(%) ↓ | Backdoor Acc(%) ↑ |
|---|---|---|---|
| Random pruning | 90.01 | 2.98 | 86.48 |
| Random reclamation | **91.10** | 81.87 | 21.44 |
| Reclamation w/ Fisher (ours) | 91.04 | **2.67** | **87.22** |
| Methods (Non-IID) | Benign Acc(%) ↑ | ASR(%) ↓ | Backdoor Acc(%) ↑ |
| Random pruning | 87.30 | 5.22 | 81.31 |
| Random reclamation | **88.52** | 88.76 | 7.40 |
| Reclamation w/ Fisher (ours) | 87.48 | **3.63** | **82.01** |

**Consensus filtering (CF).** In our design, we conduct consensus filtering to prune out the malicious/dummy parameters in the model to correct its backdoor behavior. We show in Figure 11 the performance comparison between models before/after consensus filtering. Two observations can be drawn: i) ASR is significantly lowered after consensus filtering, showing that it is necessary to remove poisoned parameters, and this is also the core reason that we can correct the backdoor behavior of a model. ii) with CF, though the model is pruned to a sparse model, its benign accuracy is higher than its dense version before pruning. The reason is that the clients are doing pruning awareness training in the training phase as enforced by Silencer. That means we are training the core sparse model to learn how to extract features of data but not the full model. The presence of other dummy/poisoned parameters that will eventually be pruned by CF would inversely interrupt the function of the core sparse model, leading to degraded benign accuracy.

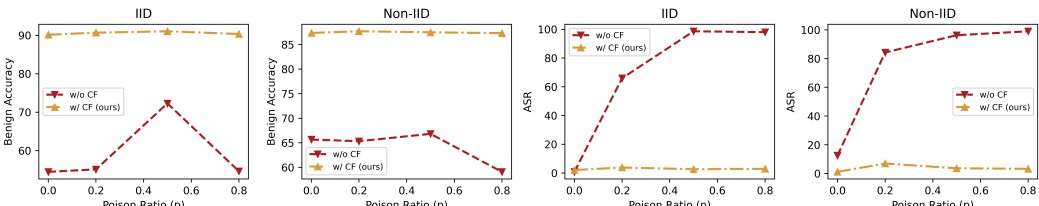

Figure 11: Impact of consensus fusion of Silencer.

### D.4 HYPER-PARAMETER SENSITIVITY ANALYSIS

We now provide the missing results of hyper-parameter sensitivity analysis. The default hyper-parameters adopted by Silencer are marked in a gray background.

**Sparsity** $s$. We set the sparsity $s$ to different values to show its impact on Silencer performance. Note that $s = 0$ reduces to vanilla FedAvg without defense. Our results are shown in Table 16, in which we can see a clear trend that: i) larger sparsity is necessary to segregate the poisoned parameters, which facilitates the later consensus filtering process. ii) Proper sparsity (e.g., 0.75) would not lead to a significant drop in benign accuracy (<1% of accuracy drop). iii) larger sparsity also brings an opportunity for applying the model acceleration technique to improve inference speed when deployed. One can choose a proper sparsity based on the given trend, considering the trade-off between robustness to attack, model performance, and inference speed.

Table 16: Performance of Silencer under different sparsity $s$.

| $s$ (IID) | Benign Acc ↑ | ASR ↓ | # of params ↓ |
|---|---|---|---|
| 0 | **91.62** | 73.41 | 6.57M |
| 0.2 | 91.50 | 55.63 | 5.26M |
| 0.5 | 91.37 | 8.37 | 3.29M |
| 0.75 | 91.04 | **2.67** | 1.64M |
| 0.9 | 88.94 | 3.12 | **0.66M** |

| $s$ (Non-IID) | Benign Acc ↑ | ASR ↓ | # of params ↓ |
|---|---|---|---|
| 0 | **89.07** | 91.36 | 6.57M |
| 0.2 | 88.78 | 83.11 | 5.26M |
| 0.5 | 88.34 | 11.81 | 3.29M |
| 0.75 | 87.48 | 3.63 | 1.64M |
| 0.9 | 84.96 | **1.34** | 0.66M |

**Initial pruning/reclamation rate** $\alpha_0$**.** In Silener, we ask the clients to evolve their masks with a decaying pruning/reclamation rate. We show in Table 17 how tuning the initial rate $\alpha_0$ impacts the practical performance. Note here that when $\alpha_0 = 0$, it reduces to sparse training over a fixed sparse network. As shown, a larger $\alpha_0$ is necessary for the defense performance of Silencer, in order to ensure the benign clients can search for their own personalized mask. But we also observe that when $\alpha_0$ is set too large, the empirical benign performance is also degraded.

**Consensus threshold** $\theta$. We show in Figure 12 the impact of consensus threshold. As shown, the chosen consensus threshold depends on the attacker number in the system. With more attackers presented, the threshold should also be set larger in order to achieve a successful defense. Another observation is that setting the threshold to a properly large number could also increase the performance

Table 17: Performance of Silencer under different initial pruning/reclamation rate $a_0$.

| $a_0$ (IID) | Benign Acc ↑ | ASR ↓ | Backdoor Acc ↑ |
|---|---|---|---|
| 1e-6 | **91.33** | 61.15 | 39.03 |
| 1e-4 | 91.04 | **2.67** | **87.22** |
| 1e-3 | 88.95 | 3.51 | 85.07 |
| 1e-2 | 87.16 | 3.25 | 84.29 |

| $a_0$ (Non-IID) | Benign Acc ↑ | ASR ↓ | Backdoor Acc ↑ |
|---|---|---|---|
| 1e-6 | **88.54** | 84.36 | 16.79 |
| 1e-4 | 87.48 | **3.63** | **82.01** |
| 1e-3 | 86.35 | 7.18 | 78.36 |
| 1e-2 | 85.99 | 7.24 | 79.29 |

of the model due to the removal of dummy/malicious parameters, most of which are not active in the benign client's pruning-aware training stage.

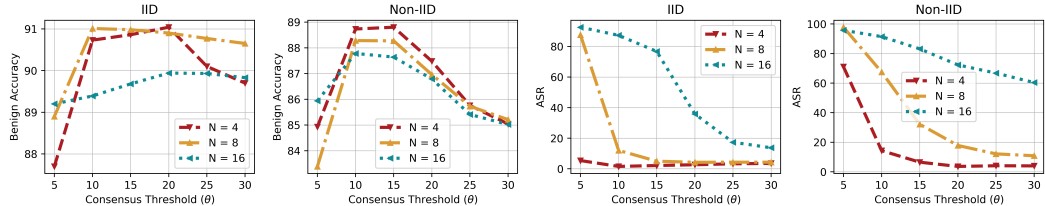

Figure 12: Impact of consensus threshold in different # of attackers setting.

### D.5 VISUALIZATION

**T-SNE visualization of representation.** Under default simulation setting, we visualize one of the benign's client model's representations before feeding into its classifier. As shown, the backdoor samples are clustered into a separate cluster for all the models trained with D-FedAvg without defense and other weaker defense baselines. This phenomenon indicates that the model can recognize the trigger and encode its information into the model representation. With this poisoned representation, the classifier can assign sufficiently large weights connecting the poisoned pattern with the target label and thereby reaching the goal of backdooring. We also observe a cluster of backdoor samples for model yielded by Silencer before the final pruning stage (i.e., Consensus filtering). But after performing stage-2 pruning, the cluster effect is canceled, i.e., the representations of backdoor samples are randomly scattered. This shows that the CF operation can indeed remove the poisoned parameters within the feature extractor and thereby perturbing the classification of the backdoor samples.

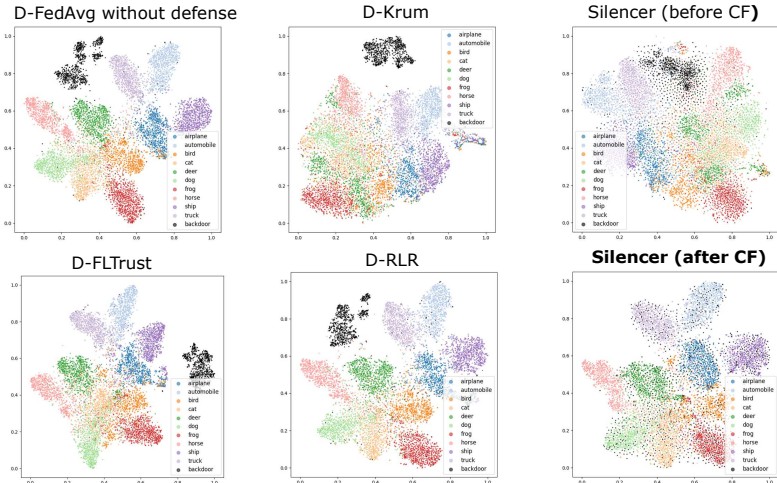

Figure 13: T-SNE of representation (of a ResNet9) before the classifier under different defenses.

**GradCam.** Under default simulation setting, we use GradCam (Selvaraju et al., 2017) to visualize the attention of the models over the original/backdoor input. As shown, the model trained by D-FedAvg

without defense or D-RLR would emphasize the backdoor trigger area when the trigger is presented but would emphasize the correct area when the trigger is not presented. In contrast, Silencer would emphasize the correct area of the input not matter when the trigger is presented or is not presented, which demonstrates that Silencer can indeed recover the model from backdoor behavior, which required special attention to associating the backdoor trigger with the target label.

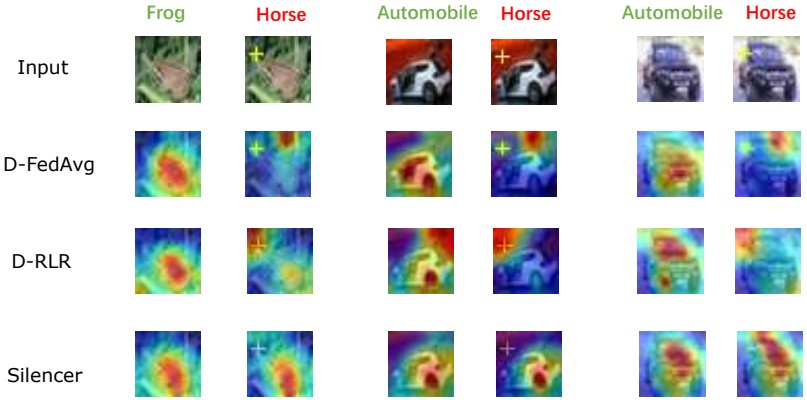

Figure 14: GradCam (of a ResNet9) under different defenses.

