# OpenReview forum: "Silencer: Pruning-aware Backdoor Defense for Decentralized Federated Learning"
_ICLR.cc/2024/Conference — ICLR 2024 Conference Withdrawn Submission_

### Official Review · Reviewer_N48q · 2023-10-18

**Soundness:** 1 poor
**Presentation:** 3 good
**Contribution:** 1 poor
**Rating:** 3
**Confidence:** 3

**Summary:**

The paper proposes Silencer, which is a model pruning scheme designed to defend against (possibly dynamic) poisoning attacks in decentralized federated learning scenarios.

The threat model assumes a small portion of malicious clients trying to minimize the loss over a poisoned dataset.
To determine which parameters in the trained architecture are important, Silencer utilizes the (approximate) fisher information (FI) metric across clients.

It uses pruning aware training utilizing FI to train local sparse models and then consensus filtering to filter globally unimportant parameters based on FI. The high-level idea (as stated in the paper) is that the poisoned parameters would only be deemed important for adversaries – namely, parameters that are shared by minority will most likely be the poisoned ones.

**Strengths:**

1. The paper is clearly written and concise.

2. Evaluation results show improvement over several previous defense techniques.

**Weaknesses:**

1. The threat model is not convincing. It assumes that the poisoned data is concentrated in a small group of clients rather than having some poisoned data that can be scattered over many clients.

2. The consensus filtering approach is not convincing. It non-explicitly assumes that malicious clients have no knowledge about any data except their own which is unrealistic. It also appears to non-explicitly assume some data similarity in non-IID setups.

3. The evaluation is insufficient. With respect to the motivation that mentions LLMs, the evaluation is based on few toy CNNs and datasets.

4. There is no sufficient evidence (theoretical or empirical) for why Silencer with DFL converges.

**Questions:**

1. Can Silencer perform well when there is a small portion of poisoned data scattered over many clients? or when malicious clients have also benign datasets?

2. How the pruning approach affects the ML performance of contemporary models? e.g., perplexity of a LLM?

3. FL is designed to keep clients' data private. With a centralized coordinator, privacy can be enhanced using DP and secure aggregation techniques. What privacy guarantees can be expected in DFL?

4. In a non-IID DFL setup, each client may have a different data distribution with different resulting FI. In that case, why consensus filtering is expected to work?

5. In a non-IID DFL setup, multiple local steps may result in bad performance without additional mechanisms to prevent client drift. Has this consideration been taken into account in the non-IID evaluation?

---

> ### Author Response · Authors · 2023-11-12
> **Authors Rebuttal (related to assumption of threat model)**
>
> Thanks for the insightful comments. We appreciate the positive comments on our wirting comments and our evaluation part. We believe the main concern of this reviewer is on the assumption of our threat model as well as our inaccurate claim on LLM in the original submission. We now try to address these issues as follows.
>
> # Assumption on  minority of attackers
> * Our method is indeed building on the assumption that attackers only constitute a minority of the clients. or formally the attacker number cannot be larger than $1/2$ of the entire number of clients.  This setting is inherited from existing data poisoning literature in FL. For example, in Krum (Blanchard el al, 2017), it is written on Page 2 that "Assuming $2f + 2 < n$, we show that Krum satisfies the resilience property aforementioned..." where $f$ is the number of malicoius clients and $n$ is the total number of clients.  A similar assumption is also made in Theorem 1 of Trimmed Mean (Yin et al., 2018). We do not relax this assumption as it serves as the foundation of our method design.
>
> Blanchard P, El Mhamdi E M, Guerraoui R, et al. Machine learning with adversaries: Byzantine tolerant gradient descent[J]. Advances in neural information processing systems, 2017, 30.
>
> Yin D, Chen Y, Kannan R, et al. Byzantine-robust distributed learning: Towards optimal statistical rates[C]//International Conference on Machine Learning. PMLR, 2018: 5650-5659.
>
> * Though we are not able to remove this assumption, it is indeed interesting to investigate how our method performs empirically under the circumstance that there is a small portion of poisoned data scattered over many clients. This use case is interesting to study because it is possible that many clients may accidentally collect the same poisoned samples from a particular location into their own dataset.
>
> We have done an experiment to verify if Silencer is robust to this attack. We will get back to you once the results are available.
>
> # Assumption that clients have no knowledge of any data except their own.
> * Our method indeed relies on this assumption. We insist that this assumption is pretty standard for a federated learning setting because the clients cannot have knowledge about other client's data given the privacy constraint.
>
> * However, we do agree that the attacker can have a completely clean dataset on their own. This dataset can be obtained by making a copy of their own data before poisoning it.  We have designed an adaptive attack based on this case.
>
> * (Experiment on new adaptive attack) Specifically, we consider another adaptive attack named BenignSearch, in which the adversary uses its own non-poisoned dataset for mask searching. Then it uses its poisoned dataset (built upon the original benign dataset) to update the parameters within the active parameters (as specified by the mask). We show how Silencer performs towards this attack under different chosen $\alpha_0$.
>
> |         Methods (IID)         | Benign Acc |  ASR  |
> |:-----------------------------:|:----------:|:-----:|
> |            DFedAvg            |    91.0    |  66.1 |
> |    Silencer($\alpha_0=0.1$)   |    91.13   |  1.83 |
> | Silencer($\alpha_0=0.01$)     | 90.52      | 4.92  |
> | Silencer($\alpha_0 = 0.001$)  | 90.80      | 3.21  |
> | Silencer($\alpha_0 = 0.0001$) | 91.21      | 65.24 |
>
>
> |       Methods (Non-IID)       | Benign Acc |  ASR  |
> |:-----------------------------:|:----------:|:-----:|
> |            DFedAvg            |    88.8    |  86.4 |
> |   Silencer($\alpha_0 = 0.1$)  |    87.70   |  9.67 |
> | Silencer($\alpha_0 = 0.01$)   | 87.37      | 44.14 |
> | Silencer($\alpha_0 = 0.001$)  | 87.43      | 37.49 |
> | Silencer($\alpha_0 = 0.0001$) | 87.49      | 68.99 |
>
> As found, Silencer can always be robust to this adaptive attack when $\alpha_0$ is set to be large enough.
>
> The reason that Silencer is robust to this adaptive attack is that those poisoned parameters will not be recognized as important for those benign clients, and therefore would not be contained in the benign masks. As a result, no matter what the attackers are going to do, their parameters will just not be recognized as useful by most of the benign clients.
>
> We hope this erases the reviewer's concern assumption on the malicious client's data.
>
>
> # Assumption on data similarity.
> We do not make any assumptions on the data similarity. In Table 6, we test Silencer's performance under different Non-IID levels, which shows its robustness to different levels of data similarity.

---

> > ### Author Response · Authors · 2023-11-12
> > **Authors Rebuttal (II)**
> >
> > # On large language model and its evaluation.
> > * We apologize for our misleading statement that "The success of large language models (LLMs) cannot be achieved without accessing a massive amount of training data.". In fact, our paper is not on the backdoor attack of LLMs but are mostly concerned with black attack in computer vision task. It basically follows the attack setting in (Gu et al., 2017). We now revise the statement as "The success of  machine learning cannot be achieved without accessing a massive amount of training data."
> >
> > Gu T, Dolan-Gavitt B, Garg S. Badnets: Identifying vulnerabilities in the machine learning model supply chain[J]. arXiv preprint arXiv:1708.06733, 2017.
> >
> > *  For evaluation on LLMs, unfortunately, we do not have enough resources to do the evaluation. Note that as our paper is mostly concerned with how to train a clean model from scratch using decentralized data,  To extend it to LLM, we may need to pre-train an at least 7B models from scratch, which needs an enormous amount of computation resources and data, and may not be affordable for us. Therefore, we try to avoid claiming that our method can be directly applied in LLMs, though it may be extended to them. We hope the reviewer can understand.
> >
> > # Lacking theoretical or empirical evidence of why Silencer with DFL converges.
> > Thanks for pointing out this issue. Indeed we do not have a formal theoretical conclusion to show that Silencer with DFL converges, but we do show in Figure 6 that the testing accuracy is stabilizing after after sufficient rounds of training.  In order to show more empirical evidence that the model is converging at least to a stationary point, we are working on printing the gradient norm of the model along the training rounds.  Stay tuned!
> >
> >
> > # Privacy guarantee in DFL.
> > We agree that a privacy guarantee by itself is very important in both FL and DFL. Differential privacy (DP) and secure aggregation are two important techniques to enhance privacy protection in FL.
> >
> > * For DP, the privacy-preserving method can be easily applied to the DFL setting by asking the clients to add DP noise for encoding their model before gossiping gradient updates.
> >
> > * For secure aggregation, it is shown in (Jeon et al, 2021) that secure aggregation can also be adapted to DFL setting but needs some necessary modification. With the Silener protocol, we may need further modification, which is an interesting research problem.  We put this into our future research agenda.
> >
> > * We have added a section in Appendix A.3 to discuss the privacy protection issue in the revised paper.
> >
> > Jeon B, Ferdous S M, Rahman M R, et al. Privacy-preserving decentralized aggregation for federated learning[C]//IEEE INFOCOM 2021-IEEE Conference on Computer Communications Workshops (INFOCOM WKSHPS). IEEE, 2021: 1-6.
> >
> > # Why Silencer work in Non-IID?
> > * Our explanation is that the poisoned parameters would not be deemed important by the benign clients, and therefore they will not be involved in their masks. This holds for both Non-IID and IID cases, and therefore will not affect the defense performance.
> > * But we do observe that the benign accuracy drops more significantly in Non-IID cases compared to the no-defense baseline. In CIFAR10, we have about 2% drop in benign accuracy for Non-IID,  but we have a smaller than 1% drop for IID cases. The explanation is that Silencer indeed leads to more divergence masks in NonIID, which results in more benign parameters being dropped in the consensus filtering process.
> >
> >
> > # About the client drift effect of multiple local steps.
> > We currently have not considered solving the client drift issue in DFL, but we do believe that considering how to integrate the existing client-drift reduction techniques, e.g,  SCAFFOLD (Karimireddy et al, 2020), Fedprox (Li et al, 2020), etc,  is important by itself. It is on our future research agenda. It is also interesting to further study how to combine existing communication reduction techniques, e.g., ACGD (Li et al, 2020), EF21 (Richtárik et al), etc, to our solutions. We have added a section in Appendix A.3 to discuss this in the paper.
> >
> > Karimireddy S P, Kale S, Mohri M, et al. Scaffold: Stochastic controlled averaging for federated learning[C]//International conference on machine learning. PMLR, 2020: 5132-5143.
> >
> > Li T, Sahu A K, Zaheer M, et al. Federated optimization in heterogeneous networks[J]. Proceedings of Machine learning and systems, 2020, 2: 429-450.
> >
> > Li Z, Kovalev D, Qian X, et al. Acceleration for compressed gradient descent in distributed and federated optimization[J]. arXiv preprint arXiv:2002.11364, 2020.
> >
> > Richtárik P, Sokolov I, Fatkhullin I. EF21: A new, simpler, theoretically better, and practically faster error feedback[J]. Advances in Neural Information Processing Systems, 2021, 34: 4384-4396.

---

> ### Author Response · Authors · 2023-11-14
> **Extra experiment results on majority attackers**
>
> As suggested by the reviewer, we conduct an experiment to test the case when every client in the system have some poisoned data (i.e., all the clients are attackers). The poison ratio of each client is set to 1%. We show the results as follows.
>
>
> |         Methods (IID)         | Benign Acc |  ASR  |
> |:-----------------------------:|:----------:|:-----:|
> |            DFedAvg            |    91.93   | 28.83 |
> |    Silencer($\alpha_0=0.1$)   |    91.38   |  7.51 |
> | Silencer($\alpha_0=0.01$)     | 90.60      | 17.69 |
> | Silencer($\alpha_0 = 0.001$)  | 91.08      | 10.10 |
> | Silencer($\alpha_0 = 0.0001$) | 91.04      | 9.74  |
>
> |       Methods (Non-IID)       | Benign Acc |  ASR  |
> |:-----------------------------:|:----------:|:-----:|
> |            DFedAvg            |    89.85   |  7.65 |
> |   Silencer($\alpha_0 = 0.1$)  |    87.73   | 45.58 |
> | Silencer($\alpha_0 = 0.01$)   | 87.70      | 23.16 |
> | Silencer($\alpha_0 = 0.001$)  | 86.89      | 17.04 |
> | Silencer($\alpha_0 = 0.0001$) | 87.45      | 27.47 |
>
> Our results show that:
> * When the poison ratio is low, even though every client in the system hosts some poisoned data and there is no defense, the ASR cannot be large enough for an effective poisoning.
> * Silencer cannot defend against attack in this scenario. In the Non-IID scenario, no matter how we tune the pruning ratio (which is a critical hyper-parameter), the ASR cannot be lowered than D-FedAvg. This is because Silencer by design is not aiming to counter the situation that every client is a potential attacker.
>
> However, we still insist that our proposed defense is valid, because in reality, it is very hard to poison every client (or larger than 50% of clients) in the system.  We will soon update the results and discussion into the revised paper. Thanks for checking!

---

> ### Author Response · Authors · 2023-11-15
> **More experiments on larger model**
>
> As suggested by the reviewers, the method should be tested with larger scale model. This issue is also mentioned by Reviewer m7hp. To erase the reviewer's concern, we test the performance of ResNet50 under CIFAR10 (default hyper-parameters in our main experiment are adopted), as follows.
>
> |   Methods (IID)   | Benign Acc |  ASR  |
> |:-----------------:|:----------:|:-----:|
> |      DFedAvg      |    91.67   | 99.99 |
> |      Silencer     |    90.85   |  9.05 |
>
> | Methods (Non-IID) | Benign Acc |  ASR  |
> |:-----------------:|:----------:|:-----:|
> |      DFedAvg      |    88.82   | 99.31 |
> |      Silencer     |    86.28   |  1.95 |
>
> * Our results show that the Larger model (ResNet50) is more vulnerable to backdoor attack if no defense is in place (with 99% of ASR). But with Silencer protocol, we still can achieve decent defense performance.
>
> We hope this erases the reviewer's concern about the chosen model for experiments. We are not able to test with a language generation task because there still lacks a clear standard of how to inject a backdoor for an LLM, although we do see some recent attacking methods in this field emerging, e.g., (Yan el al, 2023), （Xu et al, 2023), (Huang et al, 2023).
>
> Yan J, Yadav V, Li S, et al. Virtual Prompt Injection for Instruction-Tuned Large Language Models[J]. arXiv preprint arXiv:2307.16888, 2023.
>
> Xu J, Ma M D, Wang F, et al. Instructions as Backdoors: Backdoor Vulnerabilities of Instruction Tuning for Large Language Models[J]. arXiv preprint arXiv:2305.14710, 2023.
>
> Huang H, Zhao Z, Backes M, et al. Composite Backdoor Attacks Against Large Language Models[J]. arXiv preprint arXiv:2310.07676, 2023.

---

> ### Author Response · Authors · 2023-11-17
> **More experiments on convergence property**
>
> We show the client models' gradient norm over the training dataset in different rounds. The results are as follows:
>
> |   Methods (IID)   | t=0 |    t=40    | t=80 | t=120 | t=160 | t=200 | t=240 | t=280 |
> |:-----------------:|-----|:----------:|------|:-----:|-------|-------|-------|-------|
> D-FedAvg|1.604|0.03|0.021|0.019|0.014|0.011|0.009|0.007
> Silencer| 2.123|0.042|0.03|0.025|0.025|0.021|0.021|0.022
>
> |   Methods (Non-IID)   | t=0 |    t=40    | t=80 | t=120 | t=160 | t=200 | t=240 | t=280 |
> |:-----------------:|-----|:----------:|------|:-----:|-------|-------|-------|-------|
> D-FedAvg| 1.604|0.075|0.061|0.056|0.043|0.041|0.036|0.033
> Silencer| 2.123|0.117|0.092|0.109|0.087|0.097|0.085|0.096
>
> Compared with D-FedAvg, the convergence property is indeed affected in Silencer. The results seem to indicate that the iterates can only reach a neighborhood of the stationary point from the standpoint of convex/non-convex optimization.
>
> *However, we can't agree that an algorithm that cannot converge to the stationary point is a bad algorithm without any values*. For example, the generalization-aware algorithm FedSAM (Qu et al, 2022) also can only converge to a neighborhood of a stationary point, but this does not hinder it from achieving SOTA performance in the testing set.  In our paper, as we care more about robustness/practical performance on testing datasets, we insist that Silencer has its own contribution.
>
>
> Qu Z, Li X, Duan R, et al. Generalized federated learning via sharpness aware minimization[C]//International Conference on Machine Learning. PMLR, 2022: 18250-18280.

---

> ### Author Response · Authors · 2023-11-21
> **Note with appreciation**
>
> We would like to thank the reviewer for the critical comment on our paper. From your review, we think the biggest concern is about the threat model, in which we assume the attackers only constitute a minority of the clients. This threat model of course cannot capture all the scenarios and therefore is lacking in generality.
>
> However, we still want to emphasize the novelty and technical contribution of this paper.
> *  We are the first to study Fisher information as a pruning criterion to remove the poisoned parameters, and we derive two invariant patterns through an empirical study.  From the best knowledge of the authors, these findings are not available elsewhere. Note that this finding does not rely on *the minority attacker assumption*, which could facilitate future research.
> * Our pruning-aware approach Silencer relies on *the minority attacker assumption*. But we insist that the procedures we utilize, e.g., mask searching and pruning-aware training are novel among the backdoor defense literature, and should shed some light on the mechanism of how backdoor in a deep learning model is developing.
>
> * Though we did not experiment with Silencer in the modern LLMs model, we insist our work should at least bring some insights on how to mitigate backdoors in an LLM model, and therefore should be useful for future research on machine learning/LLMs.
>
> Once again, we thank the reviewer for the critical comments, and we do value them a lot. No matter it is positive or negative, we respect the final decision you made regarding our paper.

---

### Official Review · Reviewer_fqq5 · 2023-10-30

**Soundness:** 3 good
**Presentation:** 1 poor
**Contribution:** 2 fair
**Rating:** 6
**Confidence:** 3

**Summary:**

The paper propose Silencer, an algorithm resilient to poisoning attacks in gossip algorithms by identifying suspicious updates due to their diagonal of empirical Fisher information matrix and pruning the suspicious nodes at the neighbors level. The contributions are to use the empirical Fisher information matrix, to design a pruning scheme in decentralized learning, and to evaluate empirically their methods.

**Strengths:**

- Silencer maintains a good accuracy compared to other defenses mechanism
- Computing the Fisher Information matrix looks like a sound idea, and the masking strategy seems realistic
- The experiments are quite detailed, with various datasets and topology for the graphs.

**Weaknesses:**

- Silencer performances are clearly reduced as soon as the attackers try to learn the masks, making the interest of the method questionable
- The "finding" of the fact that Fisher information matrix is a good signal for different objective function seems not so novel, as per definition it is roughly the average "sensitivity" of the log-likelihood to changes of the parameters.
- page 8 is an example of excess of tables and is barely readable. There are 8 tables and 7 figures in the main text!

**Questions:**

- can you adapt your solution to accelerated gossip or asynchronous gossip ?
- could you explain the curves of the figure 1? I am not sure to see what are the conclusions of it.
- could you comment on the extra computation needed by silencer? I believe you only discussed the speedup due to sparsity, but does it compensate the extra computation needed?
- could you comment on the stability? I saw some paragraph in appendix (Decay+ Pruning Reclamation) but it is not clear to me what are the keys messages and intuition

---

> ### Author Response · Authors · 2023-11-12
> **Authors Rebuttal (I)**
>
> We thank the reviewer for pointing out the issues of our submission. Below we try to carefully address them,
>
> # Silencer performances are clearly reduced when attackers learn the masks.
>
> As shown in Table 4, our solution indeed fails to defend against Curiosity attack. However, we insist that Silencer is still robust to adaptive attacks given the following arguments.
> * It is important to note that in Curiosity attack, we assume the attackers have full knowledge of the consensus subspace, which typically cannot be obtained easily without global consensus knowledge, i.e., knowledge of others' masks in the training process.
> * Aiming to lower the probability that the attackers can infer the benign clients' masks,  we adopt a security-enhanced version of Silencer, i.e.,  Silencer-SE to mitigate the risk of mask leakage based on the idea of noise addition. We also did experiments to evaluate the performance of Silencer-SE in Table 5.
>
> # Using Fisher information as a signal for different objectives is not novel.
> * While this is not the first time using Fisher information as an indicator of the importance of the parameters, we insist that this is the first time that Fisher information is used to identify poisoned parameters in backdoor defense literature. And this is our the first empirical finding of the invariant Fisher information pattern over the backdoor attack scenario.
>
> * Moreover, the Fisher-information-based pruning-aware training should also be counted as a novel contribution. This method has not appeared in anywhere to the best knowledge of the authors.
>
> # Too many tables and figures in the main text.
>
> We apologize for the unnecessarily compact layout of the paper. We will fix this issue in our final version of the paper. Perhaps putting some experimental results in the Appendix would fix this issue.
>
> # Adaptation to accelerated gossip or asynchronous gossip.
> * Our experiment is indeed run in the accelerated gossip setting (Chen et al, 2021). in which each client would run multiple steps before sending the update for aggregation.
> *  We believe that our method can also be extended to asynchronous gossip scenarios. However, we mainly consider synchronous gossip scenarios because the asynchronous one would add complexity to our method and may potentially obscure the primary contributions.
>
> Chen Y, Yuan K, Zhang Y, et al. Accelerating gossip SGD with periodic global averaging[C]//International Conference on Machine Learning. PMLR, 2021: 1791-1802.
>
> # Explanation of the curves of the figure 1.
> In Figure 1 we report a case study by poisoning the DFL network with 10% of attackers.  The x-axis is the attack success ratio, and the y-axis demonstrates the kernel density of models within the particular range of ASR.  For example, when x=0.8, the kernel density is approximately 3.7, which is a relatively high value. This means that a large portion of the models among all the clients models having ASR 80%.  The conclusion of this figure is that DFL is vulnerable to backdoor attacks when there are a few attackers in the system. We want to demonstrate this conclusion because this is the first study of backdoor attack in DFL, despite a concurrent work  (Yar et al., 2023).
>
> # Extra computation needed by silencer and the saved computation.
>
> * The extra computation mainly comes from the maks searching process. It typically involves extracting the gradient over the local dataset and the TOPK operation of the gradient.   Among these two,  the main computation overhead comes from the gradient extraction, because we do one full pass of the client's local dataset in this stage. However, we may also use a batch of the samples to calculate the approximated gradient in order to save computation.
>
> * For sparsity in training, it currently cannot offer a practical speedup effect in our testbed. This is because we are using an unstructured sparsification technique, which cannot guarantee speed up in the current stage. However, our method is extensible to structured sparsify, which can indeed offer a practical speedup. We would like to refer to Q3.5 of a funny SNN handbook by Liu et al,2023 for addressing the concern over the practical acceleration of unstructured sparsity.
>
> Liu S, Wang Z. Ten lessons we have learned in the new" sparseland": A short handbook for sparse neural network researchers[J]. arXiv preprint arXiv:2302.02596, 2023.
>    .

---

> ### Author Response · Authors · 2023-11-12
> **Author rebuttal (II)**
>
> # Comment on  the stability of the Decay+ Pruning Reclamation
>
> * For pruning operation, we ask every client to prune out some unimportant parameters from the mask based on the Fisher information. For Reclamation, we recover the same amount of parameters that are important for the clients into the mask. The number of parameters being pruned and recovered is determined by a decaying scalar $\alpha_t$.
>
> * The motivation of the design is based on observation in Fig. (3b), which demonstrates that the FI of a benign clients is unstable and is changing between rounds. This motivates us to design an alternative pruning/recovering process to stabilize (but also be able to update) the mask, and thereby stabilize the model training process.
>
> # Warm reminder
>
> As the reviewer is mainly concerned with the presentation and some technical details of our paper, can our rebuttal clarify the reviewer's concern?  We would like to write a warm reminder that the top 25% score of ICLR submission this time is as high as 6.2. As we feel that you overall keep a supportive attitude towards of our paper, would you mind considering adjusting the score based on our feedback? Thank you again for the constructive feedback!

---

> > ### Comment · Reviewer_fqq5 · 2023-11-17
> > **Interesting answers, some points still needing clarification**
> >
> > I would like to thank the authors for their dedication during the rebuttal phase. As a lot of comments have been added, I will try to understand a bit more the answers before eventually raising my score. Here my first impressions:
> > - on BenignSearch: if I understand correctly, BenignSearch only takes into account that the attackers have the knowledge of their own data. That's great that it is does not break the algorithm, because assuming otherwise seems unrealistic. Am I mistaken? Silencer CE is more interesting, but comes to a high communication cost, so I need to think about it.
> > - on Fisher information novelty: I note that several reviewers find this point interesting and asked questions, so it seems indeed to raise interest.
> > - on the presentation: yes please do, in particular if you incorporate some clarification in the text, replacing figure by text would be great. I would nominate Figure 2 to disappear, Figure 5 to be put in appendix (maybe with addition of other topology), Table 2, 3, 6 could also move with little harm to appendix I believe.
> > - on enhanced gossip. I had a look to your new appendix A.3, that is a related work section on communication/variance reduction in gossip algorithms and the desire to let adaptions to future work. After such a big rebuttal, no one can deny that you cannot do everything. However, I would be more interested in a real discussion of the difficulty of adapting your approach that this very generic paragraph.
> > - on the computation/communication cost: not only I get the general bashing on 'you don't put the right score' but I also get an article designed to "summarize some most common confusions in SNNs, that one may come across in various scenarios such as paper review/rebuttal and talks - many drawn from the authors’ own bittersweet experiences". I guess I could apologize to be your stupid reviewer, but unfortunately reading the third "Common Confusion in SNNs" didn't answer to my very naive question: Is silencer costly in practice? Compare to other attacks, is the cost reasonable ? Even if your research implementation a bit costly, do you it could be reduced by leveraging over some structured sparsity ? I believe your answer is yes, but it is not clear to me.
> > - on stability: I note the experiments you did on the convergence for N48q.
> >
> > I will go again through the paper in the next days, and I thank you for your detailed answers

---

> ### Author Response · Authors · 2023-11-17
> **Thanks for the encouraging comment. We have more results to clarify your confusion!**
>
> # More information on BenignSearch.
> You got this.  The attackers only have the knowledge of their own data. The procedure is that the attacker maintains two copies of the local data: the first copy is their original clean dataset; the second copy is the poisoned dataset, which is obtained by modifying its clean dataset (i.e., adding backdoor trigger and modifying label for some portion of randomly chosen data). The attacker would use the clean copy for mask searching and use the poisoned copy to poison its active model parameters.
>
> # More information on Silencer-SE
> Silencer-SE is designed to enhance Silencer's robustness towards Curiosity attack. The intuition comes from the fact that Silencer cannot defend against Curiosity because of the leakage of the Consensus mask. In Silencer-SE, we add noise to lower this leakage risk.
>
> **But we need to clarify here that  Silencer-SE does not incur extra communication compared to Silencer.** In Silencer, we ask the clients to send the full dimension of model parameters to its peers, which has the same communication cost as the classical gossiping solution, e.g., D-FedAvg. This is the same for Silencer-SE, but the difference is that the model parameters are encoded before communication by adding some noise. For SIlencer-CR, we reduce communication by asking the clients to only send a subset of the parameters. However, it is not robust against adaptive attacks, as the consensus masks can be easily inferred by the attackers.
>
> # On presentation.
> Thanks for giving the concrete instructions on saving some space. We will follow that and will update the paper when rebuttal ends (modifying layout may change the order of table and figure, which may cause inconsistency and confusion over discussion, therefore we may want to postpone the update).
>
> # On adaptation to existing solution.
> We will talk about some difficulties in adapting our solution to SCAFFOLD (Karimireddy et al, 2020), FedDyn(Acar et al, 2021), etc. We envision some difficulties for these client drift correction algorithms. For example, it is challenging to deal with the centralized control variates in SCAFFOLD. Silencer enforces each client to only train on a subset of the parameters. This would incur difficulty in defining and computing control variate for drift correction. We have updated our discussion in Appendix A.3 to give specific example of the challenges.
>
> Karimireddy S P, Kale S, Mohri M, et al. Scaffold: Stochastic controlled averaging for federated learning[C]//International conference on machine learning. PMLR, 2020: 5132-5143.
>
> Acar D A E, Zhao Y, Navarro R M, et al. Federated learning based on dynamic regularization[J]. arXiv preprint arXiv:2111.04263, 2021.
>
> # On computation/communication cost.
> Silencer indeed incurs some extra computation cost, but it does not incur more communication cost. The extra computation cost comes from the mask searching process, in which we need to i) calculate the Fisher information. ii) Sort out the TOPK Fisher information. We also need to do multiple steps just like D-FedAvg (but the complexity is the same with it).  We show the clock time of these three basic procedures.
>
> | Methods (IID) | FI calculation | TOPK pruning/reclamation | Local training |
> |:-------------:|----------------|:------------------------:|----------------|
> |    DFedAvg    | 0      s        |             0     s       | 1.544    s      |
> |    Silencer   | 0.755     s     |           0.020    s      | 1.513   s       |
>
> For the FI calculation, we can save some time by using only a (or a few) batches for calculation. For the local training stage, we can potentially be faster if using structured sparsity or some more advanced unstructured sparsity technique (e.g., 2:4 sparsification). For structured sparsity, a straightforward adaptation is to ask each client to prune/recover the parameters in the granularity of the channel. For measuring the importance of each channel, we can sum the Fisher information of each coordinates within a channel. In this way, the local training cost can be reduced.
>
> #  On convergence experiment for Reviewer N48q.
> Our conclusion is that the iterates can only reach a neighborhood of the stationary point from the standpoint of convex/non-convex optimization. Just let us know if you have further comments on it. Thanks!
>
> # Straight comments from authors.
> We generally believe that not an algorithm is perfect in every aspect. We don't want to hide the weakness of our algorithm (e.g., extra computation, may fail against some adaptive attacks in some cases, cannot converge to a stationary point, cannot accelerate due to unstructured sparsity,  etc).   To clarify this, we will write a paragraph in the discussion board to highlight the weakness of our algorithms. Thank you and all the other reviewers for asking these questions, helping us thoroughly re-evaluate our algorithms.

---

> > ### Comment · Reviewer_fqq5 · 2023-11-20
> > **Raising my score**
> >
> > I thank the authors for the additional clarifications, they answered my main concerns.

---

> > > ### Author Response · Authors · 2023-11-20
> > > **Thanks for raising the score!**
> > >
> > > We would like to thank the reviewer for the re-evaluation of the Silencer paper.  We believe the main issue in our first submission is its very dense writing style, which significantly lowers the readability.  We will correct the writing layout once we got a reply from other reviewers.

---

> ### Author Response · Authors · 2023-11-22
> **On presentation**
>
> We thank the reviewer for offering advice on our presentation. As promised, we reorganized the layout of the paper by removing some unnecessary results from the main paper in our new revision. Please let us know if you think the layout is still  too compact. We will make the revision in our camera-ready version. Thank you!

---

### Official Review · Reviewer_m7hp · 2023-10-31

**Soundness:** 3 good
**Presentation:** 3 good
**Contribution:** 3 good
**Rating:** 6
**Confidence:** 3

**Summary:**

The paper demonstrates the SILENCER framework for back-door defense in the decentralized federative learning setting by pruning. The authors first stated the interesting findings when probing the weight and the Fisher information of malicious clients' weight compared with benign clients. Then based on the observation, the authors proposed a two-stage pruning-aware training by asking the client to only train a subset of the weights that are important locally and prune the weights that are considered unimportant. Based on extensive experiments, the proposed method achieves the SOTA performance.

**Strengths:**

1. The findings of the weights dynamics and the fisher information seem exciting and could benefit the following research.
2. The proposed framework achieves the SOTA performance, which is attractive.
3. The presentation of the paper is easy to understand. And there are variations of the proposed algorithm in the different settings.

**Weaknesses:**

1. Although the paper visualized the statistical comparison in Figure 3(a), it looks like the approximation is not good for coordinates around 300. Considering the evaluated models are small in experiments. I wonder if the proposed method will work on larger models, for example, ResNet50.
2. There is a gap between the observation of the stability of the Fisher information and using the magnitude of the Fisher information as the indicator for pruning.
Suppose the magnitude of the fisher information of benign models could be larger than the malicious models. In that case, I will consider this work an extension of the pruning based on the sharpness, one of the significant investigated indicators of the hessian in the generalization field. Previously, people believed that the low sharpness directly leads to better generalization. However, this is not true based on the recent paper, and I think it will comprise the contribution of the proposed method.
"A Modern Look at the Relationship between Sharpness and Generalization, ICML 2023"
3. Some major unclear content:
    a. How to see the pruning rate in Figure 3c, since the authors mentioned: ".. ASR is significantly reduced with a small pruning rate".
    b. Line 12 of algorithm 1 is different from Equation 8. I think the mask should also be included in line 12.

**Questions:**

1. What is the error when approximating the hessian with the diagonal of FI with respect to the training process of the whole model? The gradient will close to 0 when closing to convergence, but the hessian will not.
2. What is the connection between the FI stability and using the magnitude for pruning? The variance should be used to measure the stability.
3. What is the relation between the proposed method and sharpness minimization?

---

> ### Author Response · Authors · 2023-11-11
> **Authors Rebuttal (I)**
>
> Thanks for the insightful comments!  We try to address the review one-by-one as follows.
>
> # Error of statistical comparison in Figure 3(a).
> Indeed, some unignorable errors between empirical FI and Hessian are observed in Figure 3(a). This is inevitable as we are using first-order information to approximate second-order information. However, our method should be robust to this error because in our mask searching process, we are using the order of the FI, but not the exact value for pruning/reclamation.
>
>
> # On the approximation of Hessian and diagonal of FI along training
>
> * We can only conclude that the approximation will be exact if the model conditional output is equal to the data label's conditional distribution, or formally when $P$ and $Q_w$ are equal (where $P( y|x)$ is the data distribution, and $Q_w( y| x)$ is the output distribution of a model).
>
> * Indeed, the Hessian of the model will not be zero after convergence, but the fact is that empirical Fisher information would also not be zero when the model converges. Recall that the definition of empirical FI is $F_i( w) = \frac{1}{N_i} \sum_{n=1}^{N_i} \left (\nabla_{ w}  \log Q_{w}( y_n \mid  x_n) \right )^2 $, and its alternative form can be written as $F_i( w) = \frac{1}{N_i} \sum_{n=1}^{N_i} \left (\nabla_{ w}  f_i (w ; x,y )\right)^2 $, where $\nabla_{ w}  f_i (w ; x,y )$ is the gradient of cross entropy loss. Note that this term is not zero in convergence given that we only have $ \frac{1}{N_i} \sum_{n=1}^{N_i} \nabla_{ w}  f_i (w ; x,y ) =0 $ when converges.
>
>
> * Unfortunately, we did not figure out a theoretical tool to bound the error between approximation along the training process, and this is our motivation to visualize the FI pattern in Figure 3.
>
> # Connection between FI stability and using the magnitude for pruning.
> Given that the coordinates with larger diagonal FI means that these coordinates are more important. Therefore the magnitude of FI can actually reflect the poisoned parameters. We can actually combine the invariant property in Figure 3 to show that i) the poisoned parameters are unique and are shared by malicious clients. ii) the poisoned patterns are not changing accross rounds.
>  These observation indicate that:
> 1. we can safely prune those poisoned parameters based on the model's FI over one particular malicious client's data to recover the model backdoor performance.  This is corroborated by the good performance of FMP-A in Figure 3.(c).
>
> 2. It is enough to take the empirical FI over the last round model to find out poisoned parameters.  Note that empricial FI is not strictly equal to Hessian in last round, so it may not accurately reflects poisoned parameters if we don't have the observation it does not change much across round.
>
> # The relation between the proposed method and sharpness minimization?
> We would say that the relation between the proposed method and the sharpness-aware minimization is not that strong. While both methods rely on Hessian somehow, the objectives of them are totally different. Our method aims at utilizing fisher information to recover the poisoned parameters and prune them afterward, in order to improve training robustness.  On the contrary, sharpness-aware minimization is concerned with finding a flat local minima to improve generalization. The optimization methods we propose are also significantly different from sharpness-aware minimization in that we are not taking double backward operation in the local training process.  But we do appreciate the reviewer for pointing out SAM, which potentially can be combined with our method to further improve its generalization ability.
>
>
> # How to see the pruning rate in Figure 3c.
> Thanks for the reminder. We did not plot the pruning ratio in the figure, which easily leads to confusion. Now we use the size of the marker as an indicator of pruning ratio.
>
> # Line 12 of algorithm 1 is different from Equation 8. The mask should also be included in line 12.
> * Thanks for the reminder. Line 12  is different from Equation 8 because in line 10 we use an intermediate variable $w_{i,t+\frac{1}{2}}$ to capture the mask update in Equation 8. We write in this way because we would like to express how the clients are going to communicate with others in Line 11.
>
> * But we do see that there is a small typo in Line 12 because of an inconsistent definition of neighbors set. Thanks for drawing our attention to this line in the algorithm.
>
> # Warm reminder
> We hope our answer erases the concerns of the reviewers. And we would like to mention that the top 25% score of ICLR submission this time is as high as 6.2. As you mention that finding in Fisher information pruning is interesting, and it seems that you generally hold a supportive view as can be seen from your comment, would you consider raising the score based on our rebuttal? We would also appreciate further comments on related items like Fisher information, sharpness-awareness minimization, etc.  Thanks!

---

> > ### Author Response · Authors · 2023-11-15
> > **Experiment on larger models**
> >
> > As pointed out by the reviewer, it is interesting to study if the method also works for a larger model. Our initial submission is evaluated on a smaller network ResNet9 with comparable performance (>90% in CIFAR10) because training on this model save computing resource, which enables us to do more groups of experiments for verification of the algorithm's robustness.
> >
> > To erase the reviewer's concern on the model, we test the performance of ResNet50 under CIFAR10 (default hyper-parameters in our main experiment are adopted), as follows.
> >
> > |   Methods (IID)   | Benign Acc |  ASR  |
> > |:-----------------:|:----------:|:-----:|
> > |      DFedAvg      |    91.67   | 99.99 |
> > |      Silencer     |    90.85   |  9.05 |
> >
> > | Methods (Non-IID) | Benign Acc |  ASR  |
> > |:-----------------:|:----------:|:-----:|
> > |      DFedAvg      |    88.82   | 99.31 |
> > |      Silencer     |    86.28   |  1.95 |
> >
> > * Our results show that the Larger model (ResNet50) is more vulnerable to backdoor attack if no defense is in place. But with Silencer protocol, we still can achieve decent defense performance.
> >
> > We hope this erases the reviewer's concern about the robustness of our defense. We will also update the code to the anonymous codebase once we finish the rebuttal to ensure the results are objective and open to the public.

---

> ### Author Response · Authors · 2023-11-17
> **More results on Sharpness-aware minimization**
>
> We have implemented FedSAM (Qu et al. 2022) in our atttack setting. We compare D-FedSAM with Silencer (with SAM as the local optimizer) in the following table.
>
> |       Methods (IID)      | Benign Acc |  ASR  |
> |:------------------------:|------------|:-----:|
> |   DFedSAM ($\rho=0.1$)  | 90.69      | 84.07 |
> |  Silencer-SAM ($\rho=0.1$) | 88.79      |  2.87 |
> | DFedSAM($\rho=0.01$)   | 91.64      | 80.33 |
> | Silencer-SAM ($\rho=0.01$) | 91.03      | 2.72  |
>
> |     Methods (Non-IID)    | Benign Acc |  ASR  |
> |:------------------------:|------------|:-----:|
> |  DFedSAM ($\rho=0.1$)  | 86.45      | 86.03 |
> |  Silencer-SAM ($\rho=0.1$) | 81.19      |  5.40 |
> | DFedSAM ($\rho=0.01$)  | 89.10      | 88.42 |
> | Silencer-SAM ($\rho=0.01$) | 86.37      | 2.06  |
>
> * Our results show that:  i) directly applying SAM cannot guarantee good defense performance. ii) Silencer can still work in providing backdoor defense when integrating with SAM.
>
> Thanks for mentioning the angle of sharpness aware minimization.  We generally agree that Silencer and sharpness-aware minimization share some implicit relation. However, we currently cannot draw a conclusion, because there still needs further empirical study and a concrete theoretical analysis on it. But we will be more than happy to hear this reviewer's insight on it.
>
> Qu Z, Li X, Duan R, et al. Generalized federated learning via sharpness aware minimization[C]//International Conference on Machine Learning. PMLR, 2022: 18250-18280.

---

> > ### Comment · Reviewer_m7hp · 2023-11-18
> >
> > The author addressed my primary concerns. Thus, I increased my score.

---

> > > ### Author Response · Authors · 2023-11-18
> > > **Thanks for raising the score!**
> > >
> > > We sincerely thank the reviewer for raising the score. Please feel free to leave a comment if you feel something we are still not able to clarify (potentially during our interaction of other reviewers).

---

### Official Review · Reviewer_dudw · 2023-11-01

**Soundness:** 2 fair
**Presentation:** 3 good
**Contribution:** 2 fair
**Rating:** 5
**Confidence:** 3

**Summary:**

This paper presents a defense mechanism for backdoor attacks in decentralized FL based on parameter pruning. The effectiveness of the proposed method is evaluated and compared with other baseline defenses empirically.

**Strengths:**

1.	The findings on the invariance of poisoning pattern that motivates the defense is interesting.
2.	Experiments on multiple attacks, defenses, and datasets are performed.
3.	The paper is well written.

**Weaknesses:**

1.	More discussions and explanations on the invariance of poisoning pattern are needed. Why this is the case? Does this hold for particular types of backdoors or more general cases, e.g., clean and dirty label backdoors?
2.	The threat model only considers the backdoor injection during training and does not consider the possibilities of adversary manipulating the masking process, which significantly simplifies the defense design. While the authors include discussions and evaluations on adaptive attacks in the experiments, it does not exclude other malicious attacks to surpass the defense.
3.	From the experiment results, Silencer is outperformed by other defenses in some scenarios. For example, in Table 1 ASR compared with D-Bulyan, in Table 8 on the FashionMnist and GTSRB datasets.

**Questions:**

See weaknesses above.

---

> ### Author Response · Authors · 2023-11-11
> **Authors Rebuttal (I)**
>
> We thank the reviewer for the insightful comments on our paper. We address them as follows.
>
> # Why the round/client invariance pattern happens?
>
> * Our conjecture is that the backdoor trigger is a fixed and easy-to-detect pattern that can be easily classified with a few parameters within the network. This is very different from other benign patterns, which requires developing different channels (or parameters) to classify. That explains why the poisoning pattern is invariant along rounds while the benign pattern is not.
>
> *  (Why client invariance?) This is because the malicious clients share the same objective of classifying the backdoor trigger, they may develop the same poisoned parameters. However, the benign clients would exploit diversified benign patterns, and thereby the benign clients develop different patterns, but the malicious clients share the same pattern.
>
> # Does this hold for particular types of backdoors or more general cases?
> * We show that this pattern is significant for dirty label blackdoor (Gu et al., 2017) in which the attacker may add a trigger in training time and simultaneously modify the data label.
>
> * Clean-label attack is a blackdoor attack in the particular case that the attacker is not allowed to modify the label of the data.
> And it is shown by (Turner et al,2018) and (Zeng et al, 2022) that the effectiveness of this particular attack is significantly lower than the dirty-label attack, unless taking specific extra operations to recover the comparable performance, e.g., adding adversarial noise or GANs to perturb the feature of the original samples in  (Turner et al,2018) et al, or taking extra optimization on the trigger (Zeng et al, 2022). Because adversaries should have full control of their local data in FL setting, there is few motivations for them to perform clean-label blackdoor attack, which typically have lower performance than dirty-label attack. Therefore, we believe clean label attack in this field is less of interest.  However, our conjecture is that the invariant pattern also hold for clean-label attack, as the pattern actually encodes the association between the trigger and the target label.
>
> Gu T, Dolan-Gavitt B, Garg S. Badnets: Identifying vulnerabilities in the machine learning model supply chain[J]. arXiv preprint arXiv:1708.06733, 2017.
>
> Turner A, Tsipras D, Madry A. Clean-label backdoor attacks[J]. 2018.
>
> Zeng Y, Pan M, Just H A, et al. Narcissus: A practical clean-label backdoor attack with limited information[J]. arXiv preprint arXiv:2204.05255, 2022.
>
> # The threat model does not consider the case that the adversary can manipulate the masking process.
>
> Thanks for pointing out this issue in our writing of the threat model.  We actually allow the clients to manipulate the masking process and we constantly try to develop new adaptive attacks to break through our defense. As shown in our original submission, we designed two adaptive attacks:
> * i) Fix mask attack. We allow adversaries to keep their masks unchanged during the mask-searching process. We show that Silencer design can be robust to this attack.
> * ii) Curiosity attack. We assume the adversaries know the consensus mask and project their update into it. We show that Silencer is not robust to this adaptive attack. However, it is generally hard for the attackers to exactly know the consensus mask, and we do add a security-enhanced version of Silencer with Gaussian noise to mitigate the leakage of the consensus mask.
> * iii) Inspired by another reviewer, we design another adaptive attack named BenignSearch that assumes that the attacker uses a benign dataset for mask searching, but uses poisoned samples for training. We are running this experiment and release the results once it finishes.
>
>  We would also like to collect the reviewer's ideas on new adaptive attacks to break through our defense. We will test them with experiments and will release the code ASAP.
>
> # Silencer is outperformed by other defenses.
> In fact, no defense can outperform Silencer in both benign accuracy and ASR. For example, in Table 1 ASR of  D-Bulyan is lower than Silencer (3.72% vs. 10.96%), but it is at the cost of reduction of benign accuracy (80.34 vs. 87.31). This is also the case in Table 8. The ASR and benign accuracy of D-Krum and Silencer in FashionMnist is (0.25 vs 5.61) and (80.13 vs. 88.44).
>
> # Warm reminder
> We hope our answer erases the concerns of the reviewers. And we would like to mention that the top 25% score of ICLR submission this time is 6.2.  As you agree that the poisoning pattern that motivates the defense is interesting, the experiments are comprehensive, and the paper is well-written,  would you consider raising the score based on our rebuttal? We also looks forward to futher feedback on our paper. Appreciate it!

---

> > ### Author Response · Authors · 2023-11-12
> > **One more adaptive attack**
> >
> > We consider another adaptive attack named BenignSearch, in which the adversary use its own non-poisooned dataset for mask searching. Then it use its poisoned dataset (build upon the original benign dataset) to update the parameters within the active parameters (as speficied by the mask). We show how Silencer perform towards this attack under different chosen $\alpha_0$.
> > |         Methods (IID)         | Benign Acc |  ASR  |
> > |:-----------------------------:|:----------:|:-----:|
> > |            DFedAvg            |    91.0    |  66.1 |
> > |    Silencer($\alpha_0=0.1$)   |    91.13   |  1.83 |
> > | Silencer($\alpha_0=0.01$)     | 90.52      | 4.92  |
> > | Silencer($\alpha_0 = 0.001$)  | 90.80      | 3.21  |
> > | Silencer($\alpha_0 = 0.0001$) | 91.21      | 65.24 |
> >
> >
> > |       Methods (Non-IID)       | Benign Acc |  ASR  |
> > |:-----------------------------:|:----------:|:-----:|
> > |            DFedAvg            |    88.8    |  86.4 |
> > |   Silencer($\alpha_0 = 0.1$)  |    87.70   |  9.67 |
> > | Silencer($\alpha_0 = 0.01$)   | 87.37      | 44.14 |
> > | Silencer($\alpha_0 = 0.001$)  | 87.43      | 37.49 |
> > | Silencer($\alpha_0 = 0.0001$) | 87.49      | 68.99 |
> >
> > As found, Silencer can always be robust to this adpative attack when $\alpha_0$ is set to be large enough.
> >
> > The reason that Silencer is robust to this adaptive attack is that those poisoned parameters will not be recognized as important for those benign clients, and therefore would not be contained in the benign masks. Therefore, no matter what the attackers are going to do, their parameters will just not be recognized as useful by most of the benign clients.
> >
> > We hope this erases the reviewer's concern on the adaptive attack.

---

> ### Author Response · Authors · 2023-11-18
> **Warm reminder**
>
> We thank the reviewer for putting considerable effort into reviewing the paper.  We would like to confirm if our responses erase your concern.
>
> * **(Invariant pattern)** It seems that you are generally interested in our finding of round/client invariant pattern, and would like to understand what leads to this phenomenon. Does our explanation in the rebuttal make sense to you?
>
> * **(Type of backdoor attack)** For the type of backdoor attack being assumed,  we inherit the attack setting from the dirty-label backdoor literature, which is adopted in most (if not all) federated learning backdoor attack/defense, e.g., (Sun et al,2019), (Wang et al, 2020), (Ozdayi et al, 2021), etc. Defense against dirty label backdoor attacks is still an emerging area, and therefore we don't want to distract the attention of readers by defining another type of not common attack.  We hope the reviewer can understand.
>
> Sun Z, Kairouz P, Suresh A T, et al. Can you really backdoor federated learning?[J]. arXiv preprint arXiv:1911.07963, 2019.
>
> Wang H, Sreenivasan K, Rajput S, et al. Attack of the tails: Yes, you really can backdoor federated learning[J]. Advances in Neural Information Processing Systems, 2020, 33: 16070-16084.
>
> Ozdayi M S, Kantarcioglu M, Gel Y R. Defending against backdoors in federated learning with robust learning rate[C]//Proceedings of the AAAI Conference on Artificial Intelligence. 2021, 35(10): 9268-9276.
>
> * **(Adaptive attack)** To erase the reviewer's concern about adaptive attack, which assumes the attacker knows the defense mechanism and is able to modify its training/mask searching procedure to break it, we have implemented another adaptive attack  named BenignSearch. Our experimental results show Silencer is robust to this adaptive attack.  We hope this somehow erases the reviewer's concern on this issue.
>
> * **(Practical performance)**  For your concern on the practical performance, we note that other defenses can only reduce ASR to a small number *at the cost of a significant drop of benign accuracy*, which is generally not acceptable.
>
> **(Warm reminder)** Given that the rebuttal deadline is approaching,   would you mind checking if our answer solves your concern? Would you mind adjusting the score based on our rebuttal, as it is important to us?  We are also more than happy to discuss with you if you have other questions regarding our finding of invariant pattern, method design, or suggestions on new adaptive attacks, etc.  Thank you!

---

> > ### Comment · Reviewer_dudw · 2023-11-22
> >
> > I thank for the authors' efforts to resolve my concerns. However, I am still not convinced on the generality of the invariance pattern and the sufficiency of considered attacks.
> >
> > Is the invariance pattern specific to decentralized FL or also the case for FL with a central server? Can the adversary design attacks to disrupt this pattern? Is this pattern also observed for other backdoor attacks like semantic backdoor or invisible backdoor [1]?
> >
> > [1] Li, Yuezun, Yiming Li, Baoyuan Wu, Longkang Li, Ran He, and Siwei Lyu. "Invisible backdoor attack with sample-specific triggers." In Proceedings of the IEEE/CVF international conference on computer vision, pp. 16463-16472. 2021.

---

> ### Author Response · Authors · 2023-11-21
> **More results on clean-label backdoor/dirty-label backdoor**
>
> We feel that our previous answer on the Fisher pattern on dirty-label and clean-label backdoors is lacking, as we do not provide empirical evidence/in-depth research into different types of attacks. We apologize for our oversight.
>
> To better study clean-label attacks, our first step is to implement a clean-label attack in our testbed and compare between them. We use the following table to compare the ASR and benign accuracy between clean-label and dirty-label attack
>
> |   Methods (Non-IID)   | Benign Acc |  ASR  |
> |:---------------------:|------------|:-----:|
> | DFedAvg (clean label) | 89.65      | 39.66 |
> |  DFedAvg (dirty label) | 88.70      | 92.56 |
> | Silencer (clean label) | 86.71      | 3.75  |
> | Silencer (dirty label) | 87.52      | 10.99 |
>
> This result justifies that clean-label attack is less effective in federated learning, which coincides with the finding in previous literature e.g., (Turner et al,2018).
>
> To show what the Fisher pattern looks like for a clean-label backdoor attack,  we plot the results in Figure 8 in Appendix A.5 of the revised paper. Our observation is that the two invariant pattern does not hold for a clean-label backdoor attack. Our conjecture for the reason is that the poisoned parameters have not been developed well, which results in a variant FI pattern. In other words, when the data has been learned well by the model, their fisher information should be stable across rounds. When the data is similar, the stable Fisher pattern of them should be similar. This explains the invariant pattern of FI for the dirty-label backdoor attack.
>
> We would like to thank the reviewer for pointing out clean-label backdoor attack for comparison of the FI pattern, which deeper our understanding of the discovered invariant pattern.

---

> ### Author Response · Authors · 2023-11-22
> **Yes! In our last minutes! We show that the invariant patterns hold for backdoor attacks that adopt an invisible trigger.**
>
> We thank the reviewer for showing concern about the generality of the invariance pattern to us. We are able to make two conclusions based on our experiment results, which were just obtained in our last minutes before the deadline.
>
> * The invariant pattern also works for the case of FL with a central server. We simulate the classical FL scenario with a server being responsible for aggregation and show that this invariance pattern also works in this case. Please refer to Figure 7.(b) in our Appendix A.5 for our results.
>
> * The invariant pattern holds for an invisible backdoor. We implement the WaNet attack, which is an invisible backdoor blending the trigger into the semantics of the input data. Our results in Figure 7.(c) in our Appendix A.5  show that the invariant pattern of WaNet is significant in that the pattern of the malicious clients exhibits i) a strong correlation between clients within the same round, and ii) a strong correlation between different training rounds.  Therefore,  our conclusion is that  *WaNet is not able to break the invariant pattern, though it is shown in their paper that WaNet can escape human inspection, reverse-engineer-based technique (Wang et al., 2019), neuron response screening (Liuet al., 2018), and perturbation-based screening (Gao et al., 2019).
>
> **(why does invariant pattern hold?)** Our belief is that no matter what kinds of attack trigger they use, the malicious clients share the same malicious objective, and therefore their poisoning pattern should be similar. This explains why the client-level invariance should be held when the backdoor function is well-established. For round-invariance, our belief is that no matter what kind of attack is being used, the backdoor trigger is a fixed pattern that should be easily classified with a fixed group of parameters. Therefore when developing the backdoor function, the poisoned parameters are fixed in an early round of training and would not change afterward.
>
>  **(Note from authors)** As we have very limited time in preparing our latest rebuttal, we are not able to fulfill a few things that we initially planned to do, as follows:
>
>  i) We are not able to implement the invisible backdoor [1] because it requires us to train an encoder for trigger generation, which requires some time for training and testing. We therefore employ a similar invisible backdoor WaNet (Nguyen et al, 2020) for addressing the reviewer's concern.
>
> Nguyen T A, Tran A T. WaNet-Imperceptible Warping-based Backdoor Attack[C]//International Conference on Learning Representations. 2020.
>
>   ii) We plan to experiment with WaNet to see if Silencer is able to defend against it, but again we are not able to get the results due to time limitations.
>
> To compensate for this,  we want to make the following promises:
>
> **i) We will prepare all these results in the camera-ready of the paper.**
>
> **ii) The code and the checkpoints will be updated and released in our repo within a few days for public review.**
>
> All in all, we appreciate your response, which provides concrete directions for us to work on. Thank you!

---

### Author Response · Authors · 2023-11-12
**Author summary**

We thank all reviewers for making an effort to review our paper. We are excited to know that the reivewer think the finding of FI pattern is interesting (Reviewer dudw,  m7hp, fqq5), the paper is well-written and easy to understand (Reviewer dudw, m7hp, N48q), with comprehensive experiment (Reviewer dudw, m7hp, fqq5) and good results (Reviewer fqq5, N48q, m7hp), and all the reviewers seem to agree that the proposed methods are novel.

We here summarize the common issues raised by the reviewers, and because of the space limitation, we put our answers to other issues into individual responses to each reviewer.

**It is unknown if Silencer is robust to adaptive attacks when the attackers do not follow the protocol. (Reviewer dudw, N48q, fqq5).**

We understand that the reason reviewers raise this issue is because we lack a formal security analysis on how robust the defense can be. The reason is because our method design by principle does not originate from a theoretical standpoint.   To compensate for this shortcoming, we are highly motivated to conduct adaptive attacks that can potentially break through our defense. In our original submission, we have designed two adaptive attacks where the clients do not follow the mask searching process. And we added one more adaptive attack in the rebuttal, which assumes the attackers utilize a clean dataset for mask searching and project poisoned updates into its mask.  Our experimental results are as follows, which demonstrates that Silencer is also robust to this adaptive attack. We are willing to implement other adaptive attack fom the reviewers or from anyone who wants to participate into the discussion.

|         Methods (IID)         | Benign Acc |  ASR  |
|:-----------------------------:|:----------:|:-----:|
|            DFedAvg            |    91.0    |  66.1 |
|    Silencer($\alpha_0=0.1$)   |    91.13   |  1.83 |
| Silencer($\alpha_0=0.01$)     | 90.52      | 4.92  |
| Silencer($\alpha_0 = 0.001$)  | 90.80      | 3.21  |
| Silencer($\alpha_0 = 0.0001$) | 91.21      | 65.24 |


|       Methods (Non-IID)       | Benign Acc |  ASR  |
|:-----------------------------:|:----------:|:-----:|
|            DFedAvg            |    88.8    |  86.4 |
|   Silencer($\alpha_0 = 0.1$)  |    87.70   |  9.67 |
| Silencer($\alpha_0 = 0.01$)   | 87.37      | 44.14 |
| Silencer($\alpha_0 = 0.001$)  | 87.43      | 37.49 |
| Silencer($\alpha_0 = 0.0001$) | 87.49      | 68.99 |

**Can Silencer integrate other techniques like accelerated gossip/asynchronous gossip (Reviewer fqq5), DP/secure aggregation, and other optimizers for mitigating client drift (Reviewer N48q)?**

Indeed, FL is not perfect and there are a lot of issues (like communication/Non-IID/privacy/security) when it comes to real deployment. A usable algorithm should be able to counter all these issues.  Due to time limitations,  we may not be able to tackle these issues all at once during the 10-day short rebuttal period. But we do provide some discussion/reference on these existing techniques and envision how likely they are able to integrate into Silencer protocol. Please check out Appendix 3 and Appendix 4 of our revised paper, and thanks for bringing these techniques to our attention!


**Performance is not consistently good.**
>  Silencer cannot outperform some baselines in some settings (Reviewer dudw)
>  In Figure 3(a), it looks like the approximation is not good for coordinates around 300. (Reviewer m7hp)
>  There is no experiment showing that method can be used for training LLMs  (Reviewer N48q)

* For the first question, we claim that Silencer has good defense performance because its benign accuracy would not lose much while guaranteeing low ASR. Other baselines can outperform Silencer in ASR because they are losing too much benign accuracy.
* For the second question, Fisher information indeed cannot perfectly approximate Hessian, because it is based on first-order information.
* For the third question, we now remove our claim about LLM. We are mainly concerned with backdoor attacks in computer vision tasks. Though we believe the method can be extended to large language models, we do have not enough resources to experiment with LLMs.


**The presentation is unclear.**
> How to see the pruning rate in Figure 3c?  Line 12 in the algorithm is problematic? (Reviewer m7hp)
> What is the meaning of the curves of Figure 1 and its conclusion? Too many figures and tables on page 7  (Reviewer fqq5)

* We fixed (or will soon solve) these presentation issues accordingly in our revised paper.
* For the meaning of Figure 1, we plot this figure with a kernel density estimation method. The conclusion is to demonstrate that DFL is vulnerable to backdoor attacks.   Thanks again for pointing out these issues to us.